# Generative Modeling from Black-box Corruptions via Self-Consistent Stochastic Interpolants

**Chirag Modi[1]\*, Jiequn Han[2]\*, Eric Vanden-Eijnden[3,1], Joan Bruna[1,2]**
[1] New York University
[2] Flatiron Institute
[3] Machine Learning Lab, Capital Fund Management
modichirag@nyu.edu, jhan@flatironinstitute.org, {bruna,eve2}@cims.nyu.edu

## Abstract

Transport-based methods have emerged as a leading paradigm for building generative models from large, clean datasets. However, in many scientific and engineering domains, clean data are often unavailable: instead, we only observe measurements corrupted through a noisy, ill-conditioned channel. A generative model for the original data thus requires solving an inverse problem at the level of distributions. In this work, we introduce a novel approach to this task based on Stochastic Interpolants: we iteratively update a transport map between corrupted and clean data samples using only access to the corrupted dataset as well as black box access to the corruption channel. Under appropriate conditions, this iterative procedure converges towards a self-consistent transport map that effectively inverts the corruption channel, thus enabling a generative model for the clean data. We refer to the resulting method as the self-consistent stochastic interpolant (SCSI). It (i) is computationally efficient compared to variational alternatives, (ii) highly flexible, handling arbitrary nonlinear forward models with only black-box access, and (iii) enjoys theoretical guarantees. We demonstrate superior performance on inverse problems in natural image processing and scientific reconstruction, and establish convergence guarantees of the scheme under appropriate assumptions. Our source code is publicly available at https://github.com/modichirag/SCSI.

## 1 Introduction

Generative modeling has become a central aspect of high-dimensional learning. Transport-based methods, including diffusion-based models (Ho et al., 2020; Song et al., 2021) and flow-based models (Albergo & Vanden-Eijnden, 2023; Lipman et al., 2022; Liu et al., 2023b), have emerged as leading frameworks with a wide range of applications from natural image synthesis (Rombach et al., 2022) to molecular design (Watson et al., 2023). These methods rely on access to clean samples $x \sim \pi$ of the target distribution, which are plentiful in many machine learning tasks. However, in many scientific and engineering applications, such clean data of interest are unavailable. Instead, we only observe corrupted measurements $y$ through a *forward* map $y = \mathcal{F}(x)$ that is typically noisy and ill-conditioned. Examples include medical imaging (tomographic projections of internal structures), astronomical observation (atmospheric distortion), and other measurement processes subject to noise and information loss (Tarantola, 2005). As a result, the target data $x$ is never observed directly, rendering standard generative modeling inapplicable.

To further motivate the problem, consider an experiment that produces iid measurements $y_1, \ldots, y_N \in \mathcal{Y}$, each arising from some unobserved signal of interest $x_1, \ldots, x_N \in \mathcal{X}$. Scientists often have an accurate physical model, or *simulator* $\mathcal{F}$, whose output $y = \mathcal{F}(x)$ can be viewed as a sample from the conditional distribution $P(\mathrm{d}y|x)$, and the goal is to infer the signal associated with a specific measurement $\bar{y}$, i.e., to sample $P(\mathrm{d}x|y = \bar{y})$. Computing this posterior distribution requires a *prior* distribution $\pi \in \mathcal{P}(\mathcal{X})$, which is often not explicitly known. However, we can still characterize it

---

*Equal contribution.

implicitly: denoting by $\mu \in \mathcal{P}(\mathcal{Y})$ the law of the observations $y_i$, and by $\mathcal{K} : \mathcal{P}(\mathcal{X}) \to \mathcal{P}(\mathcal{Y})$ the integral operator associated with the simulator $\mathcal{F}$, i.e., $\mathcal{K}\pi = \int_{\mathcal{X}} P(\cdot|x)\pi(\mathrm{d}x)$, our desired prior distribution is the solution of the *inverse generative problem* $\mathcal{K}\pi = \mu$.

A classical framework to address this scenario is Empirical Bayes, dating back to Robbins (1992). The key idea is to consider an iterative procedure, based on the expectation-maximization (EM) algorithm: given the current 'prior' $\pi^{(k)}$, one considers the posterior $P^{(k)}(\mathrm{d}x|y) \propto P(\mathrm{d}y|x)\pi^{(k)}(\mathrm{d}x)$, and performs (approximate) posterior sampling $x_i^{(k)} \sim P^{(k)}(\cdot|y_i)$. The samples $(x_i^{(k)})_{i \le N}$ are then used to estimate the new prior $\pi^{(k+1)}$, for instance using diffusion models, as in Rozet et al. (2024); Bai et al. (2024). Despite its conceptual appeal, this Empirical Bayes approach presents two important challenges: i) one must know and be able to evaluate the likelihood term $P(\mathrm{d}y|x)$ associated with the forward model (having a simulator that merely produces samples from $P(\mathrm{d}y|x)$ does not guarantee this), and ii) the EM algorithm relies on a posterior sampling which is known to be computationally hard and whose sampling errors are generally difficult to quantify. As a result, one often needs to impose further restrictions on the forward model, such as linearity with Gaussian noise to enable Tweedie's formula for approximate posterior sampling (Daras et al., 2023; Rozet et al., 2024).

We introduce a novel framework for this inverse generative modeling task, using only a *black-box simulator* of the forward process $\mathcal{F}$. Our approach leverages stochastic interpolants (SI) (Albergo & Vanden-Eijnden, 2023; Albergo et al., 2023a) with a self-consistent training procedure: we iteratively transport observations $y$ to samples $x$ via a learned velocity field, then enforce consistency by requiring that these generated samples, when passed through $\mathcal{F}$, reproduce the original observation distribution. This scheme not only eliminates the need for clean data, but also avoids backpropagation or posterior sampling through $\mathcal{F}$, and enjoys provable convergence guarantees. As a result, our framework is applicable to nonlinear forward models (e.g., motion blur), non-differentiable operators (e.g., JPEG compression), and non-Gaussian noise (e.g., Poisson noise), substantially broadening the applicability of inverse generative modeling. Conceptually, this makes our approach akin to *model-free* reinforcement learning, which optimizes policies through interaction with a simulator, whereas most prior methods resemble *model-based* control, relying on explicit knowledge and differentiability of the underlying physics.

**Additional related works** Several recent works have aimed to generate clean data $x$ using only corrupted observations $y$. Most existing approaches, however, require the forward model to be explicitly specified and differentiable, often with additional structural assumptions. For example, Daras et al. (2023); Kawar et al. (2024); Chen et al. (2025); Zhang et al. (2025) train diffusion models with corrupted data under explicit linear forward models and additional rank condition. Akyildiz et al. (2025) learns a generative prior by directly minimizing the sliced-Wasserstein-2 distance between observed data and model outputs. There is a growing interest in generative models trained on mixtures of clean and noisy data which assume access to some clean samples (Daras et al., 2024; 2025; Lu et al., 2025; Meanti et al., 2025). While in this work we only work with corrupted observations, our approach can straightforwardly be applied to these mixed settings. On the theoretical side, Li et al. (2024; 2025) study inverse problems over measure spaces, analyzing stability, variational structures, and gradient flows. Our work complements these by introducing a practical and scalable algorithmic framework while also establishing convergence guarantees under appropriate assumptions.

Finally, concurrently and independently from our work, Hosseintabar et al. (2025) proposes an EM-based approach to inverse generative modeling using conditional diffusion models, which overcomes the aforementioned limitations of the EM algorithm. Their method can be viewed as an instance of our proposed scheme, whereby the Stochastic Interpolation is conditional, so that it directly models the posteriors $P^{(k)}(\mathrm{d}x|y)$.

## 2 PRELIMINARIES

### 2.1 STOCHASTIC INTERPOLANTS

Let $\pi$ be the clean data distribution we wish to sample from and $\mu$ be the distribution of the corrupted data that are available to us, both supported on $\mathbb{R}^d$. Following Albergo et al. (2023a), a linear stochastic interpolant $I_t$ between $\pi$ and $\mu$ is defined by

$$I_t = \alpha_t x_0 + \beta_t x_1 + \gamma_t z, \quad t \in [0, 1], \tag{1}$$

where $(x_0, x_1)$ is sampled from a joint distribution (or coupling) $\nu(\mathrm{d}x_0, \mathrm{d}x_1)$ that maintains the marginals $\int_{\mathbb{R}^d} \nu(\cdot, \mathrm{d}x_1) = \pi$, $\int_{\mathbb{R}^d} \nu(\mathrm{d}x_0, \cdot) = \mu$, and $z \sim \gamma_d$ is independent Gaussian noise. The schedules $\alpha_t, \beta_t, \gamma_t$ satisfy boundary conditions $\alpha_0 = \beta_1 = 1$, $\alpha_1 = \beta_0 = 0$, and $\gamma_0 = \gamma_1 = 0$. Define the velocity fields

$$b(t, x) := \mathbb{E}[\dot{I}_t | I_t = x] \quad , \quad g(t, x) := \mathbb{E}[z | I_t = x].$$

Then the solutions to the following reverse-time SDE

$$\mathrm{d}X_t^B = b(t, X_t^B)\mathrm{d}t + \epsilon_t \gamma_t^{-1} g(t, X_t^B)\mathrm{d}t + \sqrt{2\epsilon_t}\mathrm{d}W_t^B \tag{2}$$

have the property that if $X_{t=1}^B \sim \mu$ is independent of $W^B$, then $X_{t=0}^B \sim \pi$. Here $\epsilon_t \in C^0([0,1])$ with $\epsilon_t \geq 0$ is an arbitrary time-dependent diffusion coefficient, and $W_t^B = -W_{1-t}$. We note (2) is closely related to the SDE in score-based diffusion models: specifically, when $\gamma_t > 0$, we have $s(t, x) := \nabla_x \log \rho(t, x) = -\gamma_t^{-1} g(t, x)$, where $\rho(t, x)$ denotes the probability density of $I_t$. From now on, for simplicity, we will assume a fixed (i.e., time-independent) diffusion coefficient $\epsilon$. Note that setting $\epsilon = 0$ recovers the probability flow ODE which only depends on the field $b$. The vector fields $b$ and $g$ can be learned efficiently in practice by solving least-squares regression problems:

$$b = \arg\min_{\hat{b}} \mathcal{E}_{\pi,\mu}^b(\hat{b}) \qquad \text{with} \quad \mathcal{E}_{\pi,\mu}^b(\hat{b}) := \int_0^1 \mathbb{E}[|\hat{b}(t, I_t) - \dot{I}_t|^2]\, \mathrm{d}t\,, \tag{3}$$

$$g = \arg\min_{\hat{g}} \mathcal{E}_{\pi,\mu}^g(\hat{g}) \qquad \text{with} \quad \mathcal{E}_{\pi,\mu}^g(\hat{g}) := \int_0^1 \mathbb{E}[|\hat{g}(t, I_t) - z|^2]\, \mathrm{d}t. \tag{4}$$

where $\mathbb{E}$ denotes an expectation over the coupling $(x_0, x_1) \sim \nu$ and $z$.

Learning the velocity field $b$ and the denoiser $g$ separately allows one to generate samples via the reverse-time SDE (2) with an arbitrary diffusion coefficient schedule $\epsilon_t$. Alternatively, by fixing a specific $\epsilon_t$, one can instead learn the *combined drift* required in (2) directly through a single least-squares regression problem:

$$v = \arg\min_{\hat{v}} \int_0^1 \mathbb{E}\left[|\hat{v}(t, I_t) - \dot{I}_t - \epsilon_t \gamma_t^{-1} z|^2\right] \mathrm{d}t. \tag{5}$$

**Notation** To simplify and unify notation, we use $\Theta$ to denote the required functions for generative modeling: $\Theta = \{b\}$ in the ODE case, and $\Theta = \{b, g\}$ in the SDE case. Let $\Phi_\Theta$ denote the backward transport map induced by $\Theta$; that is, $\Phi_\Theta(y) = X_0$ under backward probability flow ODE with terminal condition $X_1 = y$ or $\Phi_\Theta(y) = X_0^B$ under reverse-time SDE (2) with terminal condition $X_1^B = y$. Accordingly, such a transport map induces a pushforward from the observation distribution $\mu$ to the clean data distribution, denoted by $\pi_\Theta := (\Phi_\Theta)_\# \mu$. Note that in the SDE case, $\Phi_\Theta$ is a random map due to the Brownian motion, and the pushforward should be interpreted as the expected pushforward, i.e., averaging over the randomness of the Brownian motion. Finally, recall that in (3), we use $\mathcal{E}_{\pi,\mu}^b(\hat{b})$ to denote the loss associated with a candidate drift function $\hat{b}$, defined with respect to the SI between $\pi$ and $\mu$. Similarly, the objective $\mathcal{E}_{\pi,\mu}^g(\hat{g})$ for the denoiser is defined in (4).

## 2.2 PROBLEM SETUP

We consider a probability distribution of interest $\pi \in \mathcal{P}(\mathcal{X})$ and a forward model $\mathcal{F} : \mathcal{X} \to \mathcal{Y}$, which we allow to be stochastic, i.e., $y = \mathcal{F}(x)$ defines a conditional distribution of $y$ given $x$. In many applications, $\mathcal{Y}$ embeds naturally into $\mathcal{X}$ without information loss, allowing us to identify the two; all numerical experiments in this paper fall into this category. For the general case, we can always construct a common state space via a lifting argument: let $\Omega := \mathcal{X} \times \mathcal{Y}$, and define $\tilde{\mathcal{F}}(x, y) := (w, \mathcal{F}(x))$, where $w$ is drawn independently from an arbitrary base measure $\pi_0 \in \mathcal{P}(\mathcal{X})$. If $\tilde{\mathcal{K}}$ denotes the integral operator associated with $\tilde{\mathcal{F}}$, then for any $\nu \in \mathcal{P}(\Omega)$, we have $\tilde{\mathcal{K}}\nu = \pi_0 \otimes \mathcal{K}\nu_\mathcal{X}$, where $\nu_\mathcal{X}$ is the marginal of $\nu$ on $\mathcal{X}$. Thus, if $\tilde{\pi} \in \mathcal{P}(\Omega)$ solves $\tilde{\mathcal{K}}\tilde{\pi} = \tilde{\mu}$ with $\tilde{\mu} = \pi_0 \otimes \mu$, then $\tilde{\pi}_\mathcal{X}$ solves the original inverse problem $\mathcal{K}\tilde{\pi}_\mathcal{X} = \mu$. In the embedding case, we identify $\mathcal{Y}$ with its image in $\mathcal{X}$ and write $\mathcal{F} : \mathcal{X} \to \mathcal{X}$; in the general case, we replace $\mathcal{X}$ with $\Omega$ and proceed analogously. Either way, we henceforth assume $\mathcal{F} : \mathcal{X} \to \mathcal{X}$.

**Injectivity at distribution level** The forward model $\mathcal{F}$ is often ill-conditioned, non-deterministic (and therefore non-invertible) as a mapping in $\mathcal{X}$, thus justifying the need to regularize the inverse problem of recovering $x$ from the observations $y = \mathcal{F}(x)$. However, the situation is different

when viewed at the level of probability measures $\mathcal{P}(\mathcal{X})$: as soon as $\mathcal{K}$ is *injective* in $\mathcal{P}(\mathcal{X})$, i.e., $\mathcal{K}\pi = \mathcal{K}\tilde{\pi}$ imples $\pi = \tilde{\pi}$, one can hope to recover $\pi$ from $\mu$ by inverting the linear relationship $\mu = \mathcal{K}\pi$. To illustrate this point, consider the white gaussian noise (AWGN) channel $y = x + \sigma\xi$, with $\xi \sim \gamma_d \equiv \mathcal{N}(0, \mathrm{I}_d)$: while the optimum reconstruction at the level of the samples (in the MSE sense) is given by the posterior mean $\hat{x} = \mathbb{E}[x|\mathcal{F}(x)]$, and generically we always have reconstruction error $\mathbb{E}\|x - \hat{x}\|^2 > 0$, the associated inverse problem at the level of distributions amounts to a deconvolution, i.e., $\mu = \pi * \gamma_\sigma$, which is invertible for any noise level $\sigma$ — and thus $\pi$ can in principle be reconstructed *exactly*. Here are a few examples of injective channels:

**Example 1.** *The AWGN Channel:* $\mathcal{F}(x) = x + \sigma\xi$, $\xi \sim \gamma_d$ *is injective for any* $\sigma < \infty$.

**Example 2** (Random Projection Channel)**.** *Let* $\mathcal{S}(d, r)$ *be the Stiefel manifold of rank* $r$ *unitary matrices in* $\mathbb{R}^d$, *and consider a distribution* $\rho \in \mathcal{P}(\mathcal{S}(d, r))$ *such that span* $\{A; A \in supp(\rho)\} = \mathbb{R}^d$. *Then the random projection channel* $\mathcal{F}(x) = (A^\top x, A) \in \mathbb{R}^r \times \mathcal{S}(d, r)$ *is injective. This channel captures applications such as tomography or inpainting.*

**Remark 1** (Channel composition preserves injectivity)**.** *We immediately verify that if* $\mathcal{F}_1$ *and* $\mathcal{F}_2$ *are two injective channels, their composition* $\mathcal{F}_1 \circ \mathcal{F}_2$ *is also injective. Therefore, channels that combine random projections and additive noise are also injective, such as Cryo-EM.*

**Restoration, Generation and Inference**    Assuming that we have sampling access to $\mu$, our goal is to find a transport map $\hat{\Phi}$ that pushes $\mu$ to a distribution $\hat{\pi} = \hat{\Phi}_\#\mu$ satisfying

$$\mathcal{K}\hat{\pi} = \mathcal{K}\hat{\Phi}_\#\mu = \mu, \tag{6}$$

i.e., $\hat{\pi}$ solves the linear inverse problem. Note that in practice our samples from $\mu$ come from a dataset of observations $\{y_i\}_i$, $y_i \sim \mu$, so our method pushes a dataset of 'corrupted' observations into another dataset $\{\hat{\Phi}(y_i)\}_i$ of samples from $\hat{\pi}$. In that sense, our model is *restoring* a dataset. In order to obtain a *bona fide* generative model for $\pi$, we thus need the additional ability to generate further independent samples. This can be easily achieved by combining our method with a standard generative model: either train a generative model for $\mu$ using the original corrupted samples $\{y_i\}_i$ and then restore its outputs via the transport map $\hat{\Phi}$, or train a generative model for $\pi$ directly on the restored dataset.

Further, the ultimate goal is often to perform posterior inference, requiring us to sample from the posterior distribution $P(dx|y)$ rather than its marginal $\pi$. Again, the natural solution is to compose our transport map with a standard conditional generative modeling procedure. Once we have sample access to the marginal $\pi$, we can produce samples from the joint distribution $(x, \mathcal{F}(x))$, which can then be used to train a conditional transport model, as in e.g., Dax et al. (2023); Zhou et al. (2024); Chen et al. (2024).

To summarize, our focus will be on the inverse problem (6) at the distribution level, since it unlocks key applications when combined with standard downstream transport-based generative procedures.

## 3    SELF-CONSISTENT STOCHASTIC INTERPOLANTS

In the standard generative modeling setting with direct access to clean data samples $x_0 \sim \pi$ and corrupted samples $x_1 \sim \mu$, one may use, for example, the independent coupling $\nu(dx_0, dx_1) = \pi(dx_0)\mu(dx_1)$ to construct a Monte Carlo approximation of the expectation in the objective (3)(4). *However, in our inverse problem setting, we only observe corrupted data from* $\mu$ *and lack access to clean samples from* $\pi$. *So it is* a priori *not obvious how to construct the SI* (1). We now describe how to construct and train a self-consistent SI using only black-box access to the forward map $\mathcal{F}$.

### 3.1    ITERATIVE SCHEME FOR SELF-CONSISTENCY

Eq (1) provides a natural transport between the observed and clean distribution, but is actionable only when one has sample access to both $\pi$ and $\mu$. Observe first that if we replace the sample access of $\mu$ by oracle access to the simulator $\mathcal{F}$, we could still build a transport from $\pi$ to $\mu$ by leveraging the fact that $\mu = \mathcal{K}\pi$. Indeed,

$$I_t = \alpha_t x + \beta_t \mathcal{F}(x) + \gamma_t z, \quad t \in [0, 1], \quad x \sim \pi, \ z \sim \gamma_d, \ x \perp z, \tag{7}$$

defines a valid interpolation between $\pi$ and $\mu$, and can be directly sampled for training the optimal vector functions $\Theta^*$. For a generic measure $\rho$ replacing $\pi$ in (7), we denote by $b_\rho$ and $g_\rho$ their associated velocity and denoiser fields.

Observe that the associated backward transport map $\Phi^* := \Phi_{\Theta^*}$ pushes observations from $\mu$ toward clean samples from $\pi$, effectively defining a *local inverse*, in the sense that $\mathcal{K}\Phi^*_\#\mu = \mu^1$. In other words, $\Phi^*$ specifies a *self-consistency* condition in the space of observation measures; see Fig. 1.

However, there is a crucial difference in our setup: rather than accessing $\{\pi, \mathcal{F}\}$, we instead have acccess to $\{\mu, \mathcal{F}\}$. The key idea is to turn the self-consistency equation $\mathcal{K}\Phi^*_\#\mu = \mu$ into a procedure that adjusts $\Theta$ to push $\mathcal{K}(\Phi_\Theta)_\#\mu$ back to $\mu$. We use SIs to connect each of these two distributions to a common 'empirical prior'[2] $\pi_\Theta := (\Phi_\Theta)_\#\mu$. Consistency is then enforced by bringing the two SIs close to each other, leading to a natural bi-level fixed-point iteration scheme; see Alg. 1. The outer loop updates $\Theta^{(k)}$ to $\Theta^{(k+1)}$ by constructing, at each step $k$, the following SI

$$I_t^{(k+1)} = \alpha_t \Phi_{\Theta^{(k)}}(y) + \beta_t \mathcal{F}(\Phi_{\Theta^{(k)}}(y)) + \gamma_t z, \quad t \in [0,1], \ y \sim \mu, \ z \sim \gamma_d, \ y \perp z. \tag{8}$$

This SI is directly sampleable given $\Theta^{(k)}$ and samples from $\mu$, and we train it using standard SI loss (3)(4) via stochastic gradient descent as the inner loop to obtain $\Theta^{(k+1)}$:

$$\Theta^{(k)} \xrightarrow[\text{'E' step}]{\text{via (8)}} I_t^{(k+1)} \xrightarrow[\text{'M' step}]{\text{minimizers in (3)(4) with } I_t^{(k+1)}} \Theta^{(k+1)}. \tag{9}$$

We remark that this bi-level scheme resembles the EM-type algorithm in Rozet et al. (2024); Bai et al. (2024), where the clean data is updated in the outer loop and the score function is retrained in the inner loop. However, their method requires an explicit linear forward model and relies on uncontrollable approximations to posterior sampling. In contrast, our data-driven backward transport map avoids these assumptions and enables learning the SI in Eq. (8) with only black-box access to $\mathcal{F}$.

We easily verify that if the forward model is injective at the distribution level, $\pi$ is the only admissible fixed point of our iterative scheme (see proof in App. A.1), similarly as the consistency guarantees in Daras et al. (2024). In Section 4 we will show that with additional assumptions beyond injectivity, one can further establish unconditional convergence guarantees for our scheme.

**Proposition 1** ($\pi$ is the only admissible fixed point). *Assume that $\mathcal{K}$ is injective and that the iterative scheme* (9) *converges to a fixed point $\Theta^*$. Then $\pi_{\Theta^*} = \pi$.*

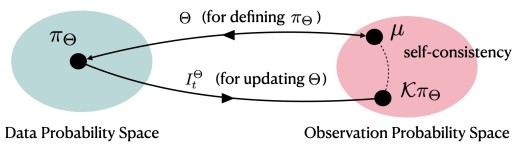

Figure 1: Schematic of the SCSI: the fixed point $\Theta^*$ satisfies $\mathcal{K}\pi_{\Theta^*} = \mu$, which in turns implies $\pi_{\Theta^*} = \pi$. No samples from $\pi$ are required—the approach only uses corrupted samples from $\mu$ and the map $\mathcal{F}$.

---

**Algorithm 1:** Training of Self-Consistent SI

1   $\Theta \leftarrow \Theta^{(0)}$       `// Initialize`
2   **for** $k$ *in* $1 \ldots K$ **do**
3      **for** $i$ *in* $1 \ldots T_{\mathrm{tr}}$ **do**
4         Sample $I_t$ in (8) with $\Theta^{(k)}$
5         SGD update of $\Theta$ via losses (3)(4)
6      $\Theta^{(k)} \leftarrow \Theta$   `// Update transport map`
7   **return** $\Theta^{(K)}$

## 3.2 TRUNCATED INNER-LOOP OPTIMIZATION FOR EFFICIENCY

For the sake of efficiency, in practice, we do not solve the inner problem to convergence at each step $k$; instead, we initialize the parameters from $\Theta^{(k)}$ and update them for $T_{\mathrm{tr}}$ gradient steps. See Alg. 1 for a description of the resulting algorithm and Alg. 2 in App. C for more details. In the special case $T_{\mathrm{tr}} = 1$, the algorithm is equivalent to treating $I_t$ in (8) as dependent on $\Theta$ but applying `stop-gradient` to $I_t$ when computing the gradient of the corresponding loss function. Nevertheless, we retain the two-loop formulation in Alg.1 to emphasize the more general form and better match the bi-level scheme introduced earlier.

---

[1] In contrast to a *global inverse*, which would require $\mathcal{K}\Phi_\#\nu = \nu$ for *all* $\nu \in \mathcal{P}(\mathcal{X})$, a much stronger condition.

[2] In the spirit of Empirical Bayes.

# 4 THEORETICAL ANALYSIS

In this seciton we analyze the iterative scheme for SCSI from Sec. 3.1. We establish convergence in KL divergence to the ground truth in the SDE setting $\epsilon > 0$, and convergence in Wasserstein in the general setting. We focus on the idealised continuous-time limit, and leave time discretization aspects for future work. In the following, we will often use a reference $L^2$ metric in the space of flows $C([0, 1] \times \mathcal{X}; \mathcal{X})$, given by $\|b\|_{\pi_{[0,1]}}^2 := \mathbb{E}_{t \sim \mathrm{Unif}[0,1]} \mathbb{E}_{x \sim \pi_t}[\|b(t, x)\|^2]$, where $\pi_t$ is the law of the oracle SI defined in (7). We define $\pi^{(k)} := (\Phi_{\Theta^{(k)}})_\# \mu$, the estimate of data distribution at (outer loop) iteration $k$. While we initially introduced the denoiser $g$, we will henceforth mainly use the score $s$ in analysis for convenience, noting that the two are equivalent.

## 4.1 CONTRACTION IN WASSERSTEIN METRIC

We first give a simple contraction bound in the Wasserstein metric, that applies both to the SDE setting (where $\epsilon > 0$) as well as the ODE setting ($\epsilon = 0$). With slight abuse of notation, given a measure $\tilde{\pi}$ and its associated $\tilde{\mu} = \mathcal{K}\tilde{\pi}$, we denote by $\Phi_{\tilde{\pi}}$ the backward transport map that pushes $\tilde{\mu}$ to $\tilde{\pi}$, obtained by integrating the SDE/ODE associated with the solution of (3, 4).

We consider the following simple assumption:

**Assumption 1** (Lipschitz Stability in $W_2$). *There exists $R > 0$ such that for any $\tilde{\pi} \in \mathcal{P}(\mathcal{X})$ we have*

$$\mathbb{E} \|\Phi_\pi - \Phi_{\tilde{\pi}}\|_\mu^2 \leq R W_2^2(\pi, \tilde{\pi}) , \qquad (10)$$

*where the expectation is with respect to the randomness of the reverse transport map.*

In words, we are assuming that the map $\tilde{\pi} \mapsto \Phi_{\tilde{\pi}}$ between $(\mathcal{P}(\mathcal{X}), W_2)$ and $L^2(\mathcal{X}; \mathcal{X}; \mu)$ is $\sqrt{R}$-Lipschitz at $\pi$. In the ODE setting, the reverse transport is a determinsitic diffeomorphism, so the expectation disappears. More generally, the Lipschitz constant $R = R(\epsilon, \gamma, \alpha, \beta)$ depends on the design of the SI, as well as the choice of diffusion coefficient during inference.

Now, since $\pi^{(k+1)} = \Phi_\#^{(k)} \mu$, we have that $(\Phi_\pi(y), \Phi_{\pi^{(k)}}(y))$, $y \sim \mu$, is a coupling between $\pi$ and $\pi^{(k+1)}$, and therefore by directly applying the Lipscthiz assumption to $\pi^{(k)}$ we immediately obtain

$$W_2^2(\pi, \pi^{(k+1)}) \leq \mathbb{E} \|\Phi_\pi(y) - \Phi_{\pi^{(k)}}(y)\|^2 \leq R\, W_2^2(\pi, \pi^{(k)}) , \qquad (11)$$

indicating that we have a contracting scheme as soon as $R < 1$. To summarize, we have

**Fact 2** (Wasserstein Linear Convergence). *Under Assumption 1, we have $W_2^2(\pi, \pi^{(k)}) \leq R^k W_2^2(\pi, \pi^{(0)})$, and therefore we have exponential convergence as soon as $R < 1$.*

We first note that the injectivity of the channel is a necessary condition to have $R < 1$. Indeed, if $\tilde{\pi}$ is such that $\mathcal{K}\tilde{\pi} = \mathcal{K}\pi = \mu$, then $(\Phi_\pi(y), \Phi_{\tilde{\pi}}(y))$, for $y \sim \mu$, is a valid coupling between $\pi$ and $\tilde{\pi}$, and therefore $W_2^2(\pi, \tilde{\pi}) \leq \mathbb{E} \|\Phi_\pi - \Phi_{\tilde{\pi}}\|_\mu^2$. In general, it is not immediate how to verify the condition $R < 1$, since it has the flavor of a 'reverse transport inequality', but in Appendix B we verify that it holds in the Gaussian AWGN setting under appropriate SNR regimes.

## 4.2 CONTRACTION IN KL DIVERGENCE

Our next result focuses on KL contraction. This contraction relies on two structural assumptions, summarised here and further described in Appendix A.2. (i) *KL Lipschitz stability*. This is the analogue of Assumption 1, where we replace the Wasserstein metric with a KL divergence: the function $\pi \mapsto f_\pi := b_\pi + \epsilon s_\pi$ is Lipschitz with respect to the KL divergence, with a Lipschitz constant of order $\epsilon$: $\forall \pi, \tilde{\pi} , \|f_\pi - f_{\tilde{\pi}}\|_{\pi_{[0,1]}}^2 \leq L\epsilon \mathrm{KL}(\pi \| \tilde{\pi})$ . (ii) *Condition Number*. We restrict the hypothesis space of target distributions $\hat{\pi} \in \mathcal{S}$ (see more details in Appendix A.2) and consider the ratio

$$\chi := \sup_{\rho \in \mathcal{S}} \frac{\mathrm{KL}(\pi \| \rho)}{\mathrm{KL}(\mathcal{K}\pi \| \mathcal{K}\rho)}. \qquad (12)$$

This condition number quantifies the injectivity of the channel, requiring regularisation in an appropriate hypothesis class. The (regularised) inverse problem becomes non-singular whenever $\chi < \infty$.

We are now ready to state the main result of this section, with the proof deferred to Appendix A.3.

**Theorem 3** (Contraction in KL). *Assume that $\epsilon > 0$ is such that (17) holds for $L_\epsilon$ and let $\chi$ be the condition number of the regularized channel. Let $\delta^{(k)} = \max(\|b^{(k)} - \hat{b}^{(k)}\|_{\pi_{[0,1]}}, \|s^{(k)} - \hat{s}^{(k)}\|_{\pi_{[0,1]}})$ be the error incurred at iteration $k$, and assume that $\delta^{(k)} \leq \delta$ for all $k$. Then, if $L_\epsilon \chi < 4$, defining $\eta = 1 + \frac{L_\epsilon}{4} - \chi^{-1}$, we have*

$$\mathrm{KL}(\pi || \pi^{(k)}) \leq 2\eta^k \mathrm{KL}(\pi || \pi^{(0)}) + O(\epsilon^{-3})\frac{\delta^2}{(1 - \sqrt{\eta})^2} \ . \tag{13}$$

Instrumental to the contraction is the ability to relate errors in measurement space back to data space — precisely what is enabled by the condition number. An interesting interpretation of Theorem 3 is that it provides *global convergence* guarantees for a seemingly complex non-convex objective function, given in (16), by replacing the ubiquitous gradient descent strategy with a tailored 'Picard-type' iterative scheme. In that sense, our guarantees go beyond the qualitative results of the self-consistency loss in Daras et al. (2024). The upper bound (13) captures the typical tradeoff between approximation and estimation errors: a 'small' function class has a smaller condition number, which improves the contraction rate, but in turn causes the error $\delta$ to increase (the proof provides explicit error dependencies in $\delta$). That said, a quantitative analysis of this tradeoff in specific function classes is beyond the scope of this work, but an interesting question deserving further attention.

**Remark 4** (Stability). *Theorem 3 shows that the scheme is stable to estimation errors of the drift and score. However, notice that the error is measured on a path distribution $(\pi_t)_{t \in [0,1]}$ different from the training distribution $(\pi_t^{(k)})_{t \in [0,1]}$, and we rely on a uniform guarantee across all iterations. In that sense, the quantity $\delta$ captures an out-of-distribution error which is more stringent than the typical Fisher stability bounds in generative diffusion literature (Chen et al., 2022; Benton et al., 2023). Finally, we remark that if one has access to an estimate $\hat{\mu}$ rather than $\mu$, the scheme pays an additional $O(\mathrm{KL}(\mu || \hat{\mu}))$ additive term, following a standard data-processing inequality argument.*

**Remark 5** (Role of Diffusion Coefficient $\epsilon$). *A key parameter controlling the strength of Theorem 3 is the diffusion coefficient $\epsilon$. On one hand, the second term in the RHS capturing learning errors is of order $O(\epsilon^{-3})$. On the other hand, the diffusion coefficients also affect the contraction rate through the Lipschitz constant $L_\epsilon$. In any case, the current KL contraction results do not cover the limiting regime of $\epsilon \to 0$ where the transport is performed with the probability flow ODE.*

**Contraction in Fokker-Plank channels**    The Lipschitz assumption (17) is admittedly difficult to verify in general. However, there is a class of channels where one can explicitly control this Lipschitz property, given by *Fokker-Plank Channels* (Wibisono et al., 2017). These are channels where the forward map $\mathcal{F}$ can be expressed as a diffusion process itself; in other words, the law of $\mathcal{F}(x)$ given $x$ agrees with the law of $X_1$, where $X_t$ solves

$$\mathrm{d}X_t = f(t, X_t)\mathrm{d}t + \sqrt{2\epsilon}\mathrm{d}W_t \ , \quad X_0 = x \ , \tag{14}$$

for some well-posed drift $f$; see also Sobieski et al. (2025) for SDEs with matrix-valued diffusion coefficients for more general inverse problems. In this case, if the Fokker-Planck representation of the channel is known, we can replace the linear SI in (7) by (14). A prominent example of a Fokker-Planck channel is the AWGN channel, where $f \equiv 0$. Now, observe that in this case the drift of the forward process does not depend on the initial distribution, thus $L = 0$. We thus obtain:

**Corollary 1** (Contraction for FP channels). *Under the same hypothesis as in Theorem 3, using the Fokker-Plank interpolant (14) yields a KL exponential contraction with rate $1 - \chi^{-1}$ for any $\epsilon > 0$.*

## 5 EXPERIMENTS

We apply self-consistent SI to a variety of forward models across three settings: (i) synthetic low-dimensional datasets, (ii) imaging tasks, and (iii) a scientific application in quasar spectral recovery. In settings (ii) and (iii), we report restored samples in comparison with the clean samples as a byproduct of the learned transport map, *though our primary goal is to learn a generative model of the clean data distribution rather than an inverse solver operating on individual corrupted samples.*

In some tasks, a latent variable $M$ associated with $\mathcal{F}$ is observed, such as the random mask accompanying each observation in the masking task. In such cases, we additionally condition the vector fields on $M$. This procedure is fully compatible with our framework: it is equivalent to appending $M$ to the observation, redefining the forward map as $\mathcal{F}(x) = (y, M)$, and keeping the corresponding channels of the SI constant with value $M$. More implementation details are provided in App. D.

## 5.1 Low Dimensional Synthetic Models

We use this setting to compare between the ODE and SDE formalism. We take the two-moon dataset for the true data distribution and consider the AWGN channel, i.e., $\mathcal{F}$ as the corruption with Gaussian noise of fixed variance $\sigma_n$. In Fig. 2, we show the results for a high noise ($\sigma_n = 1.0$) and low noise ($\sigma_n = 0.5$) setup. For low or intermediate noise, both formalisms give similar results. However, for the high noise case, ODE restoration collapses into artificially thin arms for the two moons while SDE results remain stable.

While the SDE formalism can be more robust for *highly* corrupting forward models in synthetic examples, we find that for high dimensional experiments, it is also more sensitive to hyperparameters like noise schedule and number of transport steps. On the other hand, ODE approach works well for moderate corruptions, is largely robust to choices of hyperparameters and is less computationally expensive than SDE. Hence we only present ODE results for other experiments in the main-text and defer discussion on SDE results to Appendix E.2.

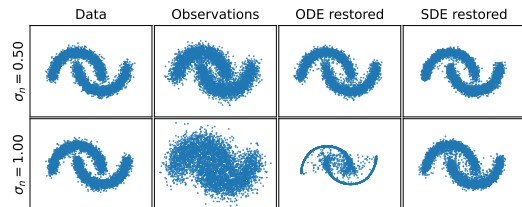

Figure 2: AWGN channel: Comparing ODE and SDE restoration for different noise levels ($\sigma_n$).

## 5.2 Imaging Tasks

**Setup** We use CIFAR-10 dataset (and CelebA dataset for JPEG compression) as the clean data distribution $\pi$ and generate one observation $y$ per image with the forward model. We model the velocity field $b$ in our SI with the U-net from Dhariwal & Nichol (2021), but using only 64 channels, resulting in ~32 million parameters[3]. We train all networks for 50,000 iterations, which required ~54 GPU hours on A100. In Appendix E.1, we provide additional ablations on how network size affects final performance.

**Comparison methods for sample restoration** We assess restoration quality by evaluating the LPIPS metric (Zhang et al., 2018) and compare our SI approach with two reference methods. The first is DPS (Chung et al., 2022), a popular and strong inverse solution based on diffusion models. DPS requires a pre-trained diffusion model on the original dataset to solve the inverse problem. Hence, we train a large diffusion model with a similar U-net architecture but 96 channels instead of 64 ($\sim 70$ million parameters), which achieved an FID of 5.16. For every inverse problem, we also did a grid search to select the best guidance strength hyperparameter as we found the good values to be very different from the recommendations in the original paper. Hence, this baseline has four advantages over our approach: i) most importantly, it *uses the clean data to train a generative model*, ii) it requires gradients of the forward map unlike our black-box only access, iii) our implementation uses a 2x larger neural network, and iv) benefits from a task-specific hyperparamter search.

The second reference is an *oracle restoration model*, obtained by training an SI model with paired clean–corrupted data as in Liu et al. (2023a). It is not a competing baseline, but an upper bound on the performance achievable by interpolant-based methods with comparable architecture and compute.

**i) Random masking** Following Daras et al. (2023); Rozet et al. (2024), this map generates an observation $y$ by masking each pixel of an image $x$ independently with probability $\rho$, and adding isotropic Gaussian noise ($\sigma_n$). As in their setting, we assume access to the mask $M$ for each $y$ and use it to condition our SI. We also pre-process the observations by adding independent standard Gaussian noise to masked pixels as it improves the final results. We show the restored images in Fig. 3a and LPIPS metric for masking probability $\rho = 0.5$ and two noise levels ($\sigma_n$) in Table 1. Our restored samples are comparable to DPS in the low-noise case and outperform it in higher-noise regimes, and they remain reasonably close to the oracle SI model across most settings. Additional quantitative metrics are provided in Appendix E.3.

To compare with inverse generative models from prior work, we use the trained SI to restore the observations; that is, we transport all observations $y$ to the data space via $\Phi_\Theta(y)$, and use these samples to train a generative diffusion model. We use the same architecture as above, but with 96 channels.

---

[3]Specifically, we use the implementation here.

Table 1: LPIPS ↓ for restoration quality of our SCSI, DPS, and the SI-Oracle. DPS requires clean pre-training data and forward-map gradients for restoration, and the oracle is trained with paired clean–corrupted data.

| Forward Model | SCSI | DPS | SI-Oracle |
|---|---|---|---|
| Random Mask ($\sigma_n = 10^{-6}$) | 0.0051 | 0.0049 | 0.0044 |
| Random Mask ($\sigma_n = 0.1$) | 0.0064 | 0.0072 | 0.0055 |
| Gaussian Blur ($\sigma_n = 0.1$) | 0.005 | 0.009 | 0.0051 |
| Gaussian Blur ($\sigma_n = 0.25$) | 0.015 | 0.025 | 0.0011 |
| Motion Blur ($\sigma_n = 10^{-6}$) | 0.0069 | 0.0026 | 0.003 |
| Motion Blur ($\sigma_n = 0.1$) | 0.011 | 0.012 | 0.003 |

Table 2: FIDs for random masking with different masking probabilities $\rho$. To account for differing architectures, 'baseline' is FID for our model on the clean CIFAR-10 data.

| Method | $\rho$ | FID ↓ |
|---|---|---|
| Ambient Diffusion | 0.20 | 11.70 |
|  | 0.40 | 18.85 |
| EM Posterior | 0.25 | 5.88 |
|  | 0.50 | 6.76 |
| **SCSI** (generated) | 0.25 | **5.38** |
|  | 0.50 | **6.74** |
| Baseline | 0.00 | 5.16 |

Table 2 shows the FID scores for observations with two different masking probabilities and negligible added Gaussian noise ($\Sigma = 10^{-6}$). SCSI vastly outperforms Ambient Diffusion (Daras et al., 2023). It is comparable with EM Posterior method (Rozet et al., 2024), but *more computationally efficient*: we required a combined $\sim 86$ GPU hours (54 and 32 GPU hours to train SI and diffusion model respectively) compared to their 512 GPU hours.

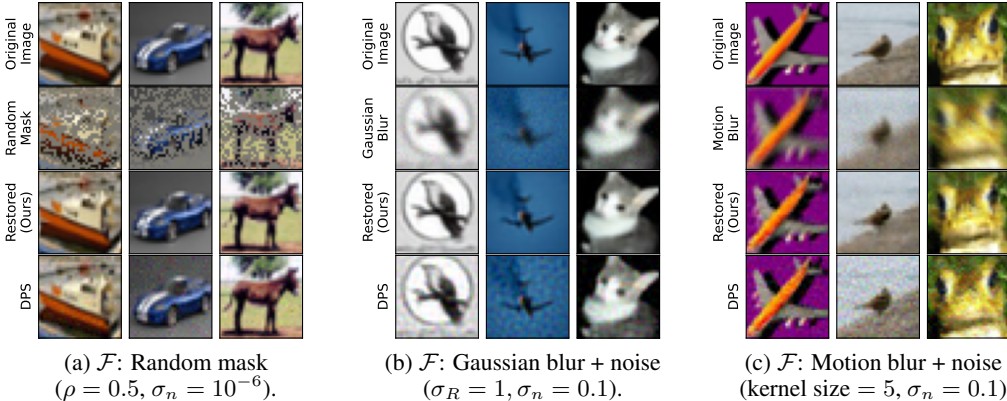

(a) $\mathcal{F}$: Random mask ($\rho = 0.5, \sigma_n = 10^{-6}$).

(b) $\mathcal{F}$: Gaussian blur + noise ($\sigma_R = 1, \sigma_n = 0.1$).

(c) $\mathcal{F}$: Motion blur + noise (kernel size = 5, $\sigma_n = 0.1$)

Figure 3: Restored samples for different forward maps from our interpolants and DPS.

**ii) Gaussian blurring with noise**    The forward map is blurring with a Gaussian kernel with $\sigma_R = 1$ and adding noise. Here we add Gaussian noises with different levels ($\sigma_n = 0.10, 0.25$) and show the results for Poisson noise case in Appendix E.6. This demonstrates that, unlike previous works, e.g., Daras et al. (2023), our approach can handle non-negligible and non-Gaussian noise.

**iii) Motion blurring**    The previous two examples involve linear forward maps. We now consider a nonlinear one: motion blur. Fig. 3c shows restored samples for observations with a 5-pixel motion kernel and small Gaussian noise ($\Sigma = 10^{-6}$). The blur direction is randomly assigned per image and assumed known for conditioning the SI. While Daras et al. (2023); Rozet et al. (2024) are limited to linear operators, SCSI handles nonlinear maps with only black-box access.

**iv) JPEG compression**    This is another common non-linear corruption operator with real-world applications. The forward map is JPEG compression with quality factor ($q$) and added Gaussian noise ($\sigma_n = 0.01$). For training, we corrupt every image randomly with a different factor $q \sim \mathcal{U}[0.1, 1]$ (where $q = 1$ implies no compression) and assume this latent parameter $q$ is known for each observation to condition the SI. Fig. 4 shows the restored image with our trained SI for different strengths of compression. The restored image gets closer to the original with lower compressions, and we restore a physically plausible image even for $q = 0.1$. In Appendix E.5.3, we show that our trained SI is stable to extrapolations of $q$ outside the training regime.

Quality : 10  Quality : 30  Quality : 50  Quality : 70  Quality : 90

Original

Figure 4: JPEG + noise: results for different compression level (Top: Corrupted; Bottom: Restored).

## 5.3 QUASAR SPECTRA

Quasars are observed through telescopes and hence the observed spectra differ from the underlying true spectra due to noise, finite spectral resolution and finite observation time. Recovering the true spectra from these observations is of interest to both study individual objects and to understand the evolution of quasars as a whole. For the true data distribution, we take the quasar spectra from Sloan Digital Sky Survey data release (Lyke et al., 2020). We isolate 30,000 quasars in redshift $z \in [2.75, 3.25]$ and consider $\lambda \in [400\mathrm{nm}, 650\mathrm{nm}]$ resulting in spectra of length $D = 1024$. We approximate the forward model with a combination of flux calibration error (offset), a Gaussian smoothing, and added Gaussian noise. Unlike imaging examples, for every observation, we randomly vary the size of the smoothing kernel within 5% and add noise with a different magnitude depending on a randomly chosen SNR. We assume we do not have access to these latent parameters for any observation. In Fig. 5, we show the restored spectra for observations in the two extreme regimes that different telescopes operate in: i) observations with high spectral resolution and low SNR (high noise), and ii) those with low spectral resolution and high SNR. For comparison, we also show Wiener filter (WF) restoration as a baseline. SCSI is best able to recover important features like peak heights and locations which are used to determine the distance and metallic composition of quasars.

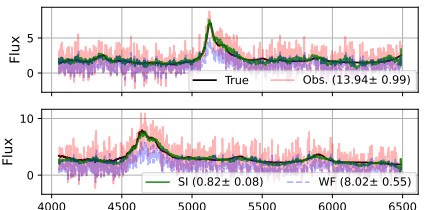
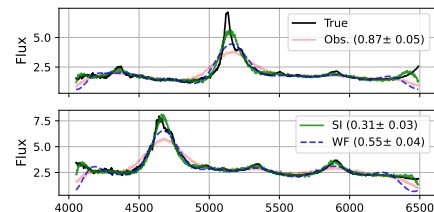

Figure 5: Restored quasar spectra for different scenarios. Legends show MSE over 1000 samples. **Left:** high spectral resolution, low SNR; $\Delta \log \lambda = \frac{1}{5000} \pm 5\%$, SNR $\in [0.5, 2]$. **Right:** low spectral resolution, high SNR; $\Delta \log \lambda = \frac{1}{50} \pm 5\%$, SNR $\in [30, 50]$.

## 6 CONCLUSION

We presented a self-consistent SI framework for reconstructing the underlying data distribution using only corrupted observations and a black-box forward model. The proposed bi-level iterative scheme is computationally practical and enjoys provable convergence under suitable assumptions. Compared to existing approaches, the proposed SCSI accommodates a much broader class of nonlinear and non-Gaussian forward models. Experimentally, we demonstrated its effectiveness across a range of inverse problems, achieving competitive performance even against methods that rely on additional access to the forward model (e.g., Ambient Diffusion) or even clean data (e.g., DPS). Interestingly, our theoretical analysis as well as our experiments point towards an advantage of the ODE-based transport over the SDE counterpart.

Looking ahead, with recent extensions of stochastic interpolants to discrete domains (Gat et al., 2024; Kim et al., 2025), our approach can also be adapted to corrupted discrete data. In addition, an important aspect to be further investigated is the comparison between marginal and posterior transport, as in the EM algorithm and its diffusion-based implementations (Rozet et al., 2024; Hosseintabar et al., 2025) – both captured in our framework by adapting the SI; see Appendix C.2. While the EM scheme is convergent under weaker assumptions than the marginal transport we analyzed in Section 4, the analysis in Appendix B demonstrates (on simple settings such as Gaussian AWGN) an inherent advantage of marginal transport in convergence rates, particularly for the probability flow ODE.

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

# A    PROOFS

For notational simplicity, we use $\mathcal{K}$ to denote $\mathcal{K}_{\mathcal{F}}$, the forward integral operator that pushes $\pi$ to $\mu$, and we adopt this shorthand throughout this section without risk of ambiguity.

## A.1    PROOF OF PROPOSITION 1

*Proof.* Recall our iterative scheme as

$$\Theta^{(k)} \xrightarrow[\text{`E' step}]{\text{via (8)}} I_t^{(k+1)} \xrightarrow[\text{`M' step}]{\text{minimizers in (3)(4) with } I_t^{(k+1)}} \Theta^{(k+1)}. \tag{15}$$

If the above iteration converges to a fixed point $\Theta^*$, and the channel is injective at the level of distributions, i.e., $\mathcal{K}\tilde{\pi} = \mathcal{K}\pi$ implies $\tilde{\pi} = \pi$, then the corresponding transport map $\Phi_{\Theta^*}$ transports corrupted samples from $\mu$ into clean samples from $\pi$. To see this, consider $\pi_{\Theta^*} := (\Phi_{\Theta^*})_\# \mu$. We prove only the SDE case, as the ODE case corresponds to the special case when $\epsilon = 0$. By definition of $\pi_{\Theta^*}$ and the property of time-reversal SDE, $\Theta^*$ transports $\pi_{\Theta^*}$ to $\mu$ under the forward SDE (Anderson, 1982; Song et al., 2021)

$$\mathrm{d}X_t^F = b(t, X_t^F)\mathrm{d}t - \epsilon_t g(t, X_t^F)\mathrm{d}t + \sqrt{2\epsilon_t}\mathrm{d}W_t.$$

On the other hand, since $\Theta^*$ is the optimal solution trained from the SI between $\pi_{\Theta^*}$ and $\mathcal{K}\pi_{\Theta^*}$, the above forward SDE also transport samples from $\pi_{\Theta^*}$ to $\mathcal{K}\pi_{\Theta^*}$ (Albergo et al., 2023a). As a result we must have $\mathcal{K}_{\mathcal{F}}\pi_{\Theta^*} = \mu$, which means that $\pi_{\Theta^*} = \pi$ thanks to injectivity. ∎

**Loss function perspective**    Our iterative scheme can be viewed as a specific procedure to find a fixed point $\Theta^*$ satisfying self-consistency. Alternatively, such a fixed point can be characterized as a minimizer of a loss function that penalizes discrepancies between two transport descriptions. Given a generic pair $\Theta = \{b, g\}$ of drift and denoiser models, the corresponding backward transport defines a distribution $\pi_\Theta := (\Phi_\Theta)_\# \mu$, and then the objectives associated with the SI between $\pi_\Theta$ and $\mathcal{K}\pi_\Theta$ defines minimizers $b_{\pi_\Theta}$ and $g_{\pi_\Theta}$. We seek to align them with the original pair via the loss

$$\mathcal{L}(b, g) = \|b - b_{\pi_{\{b,g\}}}\|^2 + \|g - g_{\pi_{\{b,g\}}}\|^2, \tag{16}$$

where $\|\cdot\|$ here denotes an $L^2$ with respect to an arbitrary base measure. The main challenge when analyzing gradient-based optimization of this loss is the highly non-linear dependencies arising from the transport map.

## A.2    ASSUMPTIONS FOR THE KL CONTRACTION

In this subsection we introduce our assumptions required to establish linear convergence rates for the KL divergence.

**KL Lipschitz stability**    Recall our definition of $b_\pi$, $s_\pi$, the velocity and score associated with the SI in (7). To control the contraction along iterations, we will assume that the function $\pi \mapsto f_\pi := b_\pi + \epsilon s_\pi$, that maps a given data model $\pi$ to the drift of a Fokker-Plank equation transporting $\pi$ to $\mathcal{K}\pi$, is Lipschitz with respect to the KL divergence, with a Lipschitz constant of order $\epsilon$:

**Assumption 2** (Lipscthiz Stability in KL)**.**

$$\forall \pi, \tilde{\pi} , \ \|f_\pi - f_{\tilde{\pi}}\|_{\pi_{[0,1]}}^2 \leq L\epsilon \mathrm{KL}(\pi||\tilde{\pi}) . \tag{17}$$

In words, a SI builds a diffusion bridge between $\pi$ and $\mathcal{K}\pi$, and $L\epsilon$ measures the sensitivity of its drift to initial conditions. To avoid confusion we shall write $L = L_\epsilon$ to emphasize that this Lipscthiz constant does depend on the diffusion coefficient. Note that $\|f_\pi - f_{\tilde{\pi}}\|^2 \leq 2\|b_\pi - b_{\tilde{\pi}}\|^2 + 2\epsilon^2\|s_\pi - s_{\tilde{\pi}}\|^2$, so the assumption balances the Lipschitz smoothness of the drift $b$ (of order $O_\epsilon(1)$) and the score $s$ (of order $O(\epsilon^2)$) via the diffusion coefficient $\epsilon$. For general channels, this assumption indirectly imposes a macroscopic diffusion coefficient $\epsilon = \Theta(1)$. Indeed, in the limit of infinitesimally small diffusion $\epsilon \to 0$ (capturing the ODE inference setting), this assumption imposes $\pi \mapsto b_\pi$ to be *constant* (ie, $b_\pi$ is independent of $\pi$), and therefore that the channel $\mathcal{F}$ is a deterministic diffeomorphism $\mathcal{T} : \mathcal{X} \to \mathcal{X}$, whose ODE representation is given by $b$. Notice that the Lipschitz constant depends on the SI design, and might be reduced by going beyond linear interpolants and more effectively 'preconditioning' to $\mathcal{K}$, which we leave for future work.

**Condition number**    An important aspect of the problem is that there are two distinct notions of error, whether it is measured on the 'data' side, i.e., $\mathrm{KL}(\pi||\pi^{(k)})$, or on the 'observation' side, i.e., $\mathrm{KL}(\mu||\mu^{(k)}) = \mathrm{KL}(\mathcal{K}\pi||\mathcal{K}\pi^{(k)})$. Since the learner only has access to data from $\mu$, a necessary condition to guarantee that we can recover the original data distribution is *injectivity*, i.e., that $\mathrm{KL}(\mathcal{K}\pi||\mathcal{K}\hat{\pi}) = 0$ implies $\pi = \hat{\pi}$. However, this is not sufficient to provide a quantitative estimate of $\mathrm{KL}(\pi||\hat{\pi})$ in terms of $\mathrm{KL}(\mu||\mathcal{K}\hat{\pi})$. In other words, the inverse problem $\mathcal{K}\pi = \mu$ is generally singular in $\mathcal{P}(\mathcal{X})$, even for the simplest channels, due to the infinite-dimensional nature of the domain.

To mitigate this issue, we need to regularize this inverse problem by restricting (or penalizing) the domain of possible velocities and scores arising from the SI objectives (3)(4), so that the resulting constrained optimization returns $\hat{b}_{\pi^{(k)}}, \hat{s}_{\pi^{(k)}} \in \mathcal{B}_\lambda$, where $\mathcal{B}_\lambda$ is indexed by a complexity measure $\lambda$, e.g., neural networks with $O(\lambda^{-1})$ parameters. In turn, these regularised objectives inject regularity in $\pi^{(k)}$, i.e., for all $k$ we have $\pi^{(k)} \in \mathcal{S}_\lambda$, the class of terminal densities obtained by running a Fokker-Plank equation with drifts in $\mathcal{B}_\lambda$. We can now consider the *condition number* of $\mathcal{K}$ around $\pi$:

$$\chi := \sup_{\rho \in \mathcal{S}_\lambda} \frac{\mathrm{KL}(\pi||\rho)}{\mathrm{KL}(\mathcal{K}\pi||\mathcal{K}\rho)} \ . \tag{18}$$

Note that by the data-processing inequality, we always have $\chi \geq 1$. The (regularised) inverse problem becomes non-singular whenever $\chi < \infty$. The purpose of regularisation, in this abstract context, is to restrict the range $\mathcal{S}_\lambda$ as to make $\chi$ small, while maintaining a small approximation error. Observe that, if $\rho \in \mathcal{S}_\lambda$, by Girsanov's theorem we have $\mathrm{KL}(\pi||\rho) \leq \epsilon^{-1}\|b^* - \hat{b}_\rho\|^2 + \epsilon\|s^* - \hat{s}_\rho\|^2 < \infty$, which shows that $\chi$ is well-defined.

A particularly simple form of regularisation is to consider a continuous parametric model $\{b_\omega, s_\omega\}$ where $\omega \in \mathcal{D}$ is in a *compact* domain, which encompasses most practical setups. Combined with the injectivity of the channel, this allows us to have $\chi < \infty$. For technical reasons, we consider the following misspecified setting. Note that, under these general conditions, unsurprisingly, we are unable to quantify the condition number.

**Proposition 2** (Finite condition number for compact hypothesis class). *Assume that $\mathcal{K}$ is injective, that $\mathcal{D}$ is a compact parameter space, with continuous parametrization of the drift and score models, and that $\pi$ cannot be exactly represented by the model. Then $\chi < \infty$.*

*Proof.* Let $F : \mathcal{D} \to \mathcal{P}(\mathcal{X})$ be the function that maps a model $\{b_\Theta, s_\Theta\}$ to $F(\Theta) = \pi_1$, where $(\pi_t)_t$ is the marginal law of $(X_t)_t$, which solves the SDE

$$dX_t = (b_\theta(t, X_t) + 2\epsilon s_\theta(t, X_t))dt + \sqrt{2\epsilon}dW_t \ , \tag{19}$$

$$X_0 \sim \mu \ . \tag{20}$$

Define $G(\Theta) := \mathrm{KL}(\mathcal{K}\pi||\mathcal{K}F(\Theta))$. We claim that $G$ is positive for all $\Theta \in \mathcal{D}$ and that $G$ is lower semi-continuous. Indeed, since we are assuming a misspecified model, we have $\mathrm{KL}(\pi||F(\Theta)) > 0$ for all $\Theta \in \mathcal{D}$, which implies $G(\Theta) > 0$ for all $\Theta \in \mathcal{D}$ thanks to the injectivity of $\mathcal{K}$.

Moreover, the mapping $\nu \mapsto \mathrm{KL}(\mu||\nu)$ is lower semi-continuous in the weak topology. This follows from the Donsker-Varadhan variational representation of the KL divergence:

$$\mathrm{KL}(\mu||\nu) = \sup_{f \in \mathcal{C}_b} \left\{ \langle f, \mu \rangle - \log\langle e^f, \nu \rangle \right\} \ .$$

The map $\nu \mapsto -\log\langle e^f, \nu \rangle$ is weakly continuous for all $f \in \mathcal{C}_b$, and the supremum of continuous functions is lower semicontinuous. Now, consider any sequence $(\Theta_n)_n$ such that $\|\Theta_n - \Theta\| \to 0$ as $n \to \infty$. By Girsanov's theorem, observe that

$$\mathrm{KL}(F(\Theta)||F(\Theta_n)) \leq \epsilon^{-1}\|b_\Theta - b_{\Theta_n}\|^2 + \epsilon\|s_\Theta - s_{\Theta_n}\|^2 \ , \tag{21}$$

which shows that $\mathrm{KL}(F(\Theta)||F(\Theta_n)) \to 0$ as $n \to \infty$ thanks to the continuity of the mappings $\Theta \mapsto \{b, g\}_\Theta$. By Pinsker's inequality, we also have that $\|F(\Theta) - F(\Theta_n)\|_{\mathrm{TV}} \to 0$, which shows that $F(\Theta_n)$ converges weakly to $F(\Theta)$, and therefore

$$\liminf_{n \to \infty} G(\Theta_n) \geq G(\Theta) \ , \tag{22}$$

showing that $G$ is LSC as claimed.

Now, observe that

$$\mathrm{KL}(\pi||F(\Theta)) \le \epsilon^{-1}\|b_\Theta - b^*\|^2 + \epsilon\|s_\Theta - s^*\|^2 := J(\Theta) \,,$$

and

$$\frac{\mathrm{KL}(\pi||F(\Theta))}{\mathrm{KL}(\mu||\mathcal{K}F(\Theta))} \le \frac{J(\Theta)}{G(\Theta)} := r(\Theta) \,. \tag{23}$$

The function $r$ is the ratio between a continuous function and a positive, lower semicontinuous function. It follows that $r$ is upper semi-continuous, and therefore

$$\chi \le \sup_{\Theta \in \mathcal{D}} r(\Theta) < \infty \,,$$

since USC functions attain a maximum over compact sets. ∎

## A.3 PROOF OF THEOREM 3

*Proof.* The strategy of the proof is to establish a comparison between $\mathrm{KL}(\pi||\pi^{(k)})$ and $\mathrm{KL}(\pi||\pi^{(k+1)})$ by exploiting the relationship between the diffusion bridges that relate them.

For that purpose, let $I_t^*$ be the *oracle* SI, given by

$$I_t^* = \alpha_t X + \beta_t \mathcal{F}(X) + \gamma_t z \,, \quad X \sim \pi^*. \tag{24}$$

Let $\pi_t$ be the law of $I_t^*$. It solves the Fokker-Planck equation

$$\partial_t \pi_t = \nabla \cdot ((-b^* - \epsilon s^*)\pi_t) + \epsilon \Delta \pi_t \,, \tag{25}$$
$$\pi_0 = \pi \,, \ \pi_1 = \mathcal{K}\pi = \mu \,,$$

where

$$b^*(t,x) := \mathbb{E}[\dot{I}_t^* \,|\, I_t^* = x] \,, \tag{26}$$
$$s^*(t,x) := -\mathbb{E}[\gamma_t^{-1}z \,|\, I_t^* = x] \,,$$

as well as the reverse Fokker-Planck equation

$$\partial_t \pi_t = \nabla \cdot ((-b^* + \epsilon s^*)\pi_t) - \epsilon \Delta \pi_t \,, \tag{27}$$
$$\pi_1 = \mathcal{K}\pi = \mu \,, \ \pi_0 = \pi \,.$$

Consider also the SI at iteration $k$ of our algorithm. Given $\pi^{(k)}$, we consider the interpolant

$$I_t^{(k)} = \alpha_t X + \beta_t \mathcal{F}(X) + \gamma_t z \,, \quad X \sim \pi^{(k)} \,, \tag{28}$$

its associated (exact) drift and scores

$$b^{(k)}(t,x) := \mathbb{E}[\dot{I}_t^{(k)} \,|\, I_t^{(k)} = x] \,, \tag{29}$$
$$s^{(k)}(t,x) := -\mathbb{E}[\gamma_t^{-1}z \,|\, I_t^{(k)} = x] \,,$$

as well as the estimated drifts and scores, that we recall are given by

$$\hat{b}^{(k)} = \arg\min_{\hat{b}} \mathcal{E}_{\pi^{(k)},\mathcal{K}\pi^{(k)}}^b(\hat{b}) + \lambda\mathcal{R}(\hat{b}) \,, \ \hat{s}^{(k)} \arg\min_{\hat{s}} \mathcal{E}_{\pi^{(k)},\mathcal{K}\pi^{(k)}}^s(\hat{s}) + \lambda\mathcal{R}(\hat{s}) \,. \tag{30}$$

They define respectively a forward Fokker-Planck equation

$$\partial_t \pi_t = \nabla \cdot ((-b^{(k)} - \epsilon s^{(k)})\pi_t) + \epsilon \Delta \pi_t \,, \tag{31}$$
$$\pi_0 = \pi^{(k)} \,, \ \pi_1 = \mathcal{K}\pi^{(k)} = \mu^{(k)} \,,$$

and a reverse Fokker-Planck equation (notice its initial condition is $\mu$):

$$\partial_t \pi_t = \nabla \cdot ((-\hat{b}^{(k)} + \epsilon \hat{s}^{(k)})\pi_t) - \epsilon \Delta \pi_t \,, \tag{32}$$
$$\pi_1 = \mu \,, \ \pi_0 := \pi^{(k+1)} \,.$$

It is also useful to define $f := b + \epsilon s$ to be the total drift of the forward (i.e., from data to measurements) diffusion; with the corresponding oracle $f^*$, iterate $f^{(k)}$ and estimated $\hat{f}^{(k)}$ versions defined analogously. From (25), (27), (31) and (32) we immediately verify that the reverse drift becomes $-f + 2\epsilon s$.

The following lemma relates the rate of KL along two SDEs. We reproduce the proof later for completeness, but it is a known result, e.g., (Boffi & Vanden-Eijnden, 2023, Proposition 1) or (Albergo et al., 2023a, Lemma 2.22):

**Lemma 1** (KL divergence along two diffusion processes). *Let $dX_t = b(t, X_t)dt + \sqrt{2\sigma}dW_t$ and $dY_t = a(t, Y_t)dt + \sqrt{2\sigma}dW_t$ be two diffusions, and $\mu_t$, $\nu_t$ denote the marginal law of $X_t$ and $Y_t$ respectively. Then*

$$\frac{d}{dt}\mathrm{KL}(\mu_t||\nu_t) = -\sigma\mathrm{I}(\mu_t||\nu_t) + \mathbb{E}_{\mu_t}\langle b - a, \nabla\log\mu_t - \nabla\log\nu_t\rangle , \tag{33}$$

*where $\mathrm{I}(\mu||\nu) = \mathbb{E}_\mu[\|\nabla\log\mu - \nabla\log\nu\|^2]$ is the Fisher divergence.*

In the particular setting where $b = a$, one obtains a de Bruijn identity:

**Lemma 2** (de Bruijn Identity).

$$\frac{d}{dt}\mathrm{KL}(\pi_t||\pi_t^{(k)}) = -\sigma\mathrm{I}(\pi_t||\pi_t^{(k)}) . \tag{34}$$

Besides a control of the marginal KL, we will also use Girsanov's theorem to obtain control of the KL divergence between path measures of $(X_t)_t$ and $(Y_t)_t$:

**Lemma 3** (Girsanov Theorem). *Let $dX_t = b(t, X_t)dt + \sqrt{2\sigma}dW_t$ and $dY_t = a(t, Y_t)dt + \sqrt{2\sigma}dW_t$ be two diffusions, and let $\mu_{[0,T]}$ and $\nu_{[0,T]}$ be the path measures of $X_t$ and $Y_t$, respectively. Assume the Novikov integrability condition. Then*

$$\mathrm{KL}(\mu_{[0,T]}||\nu_{[0,T]}) = \mathrm{KL}(\mu_0||\nu_0) + \frac{1}{4\sigma}\mathbb{E}_{\mu_{[0,T]}}\int_0^T \|a(t, x) - b(t, x)\|^2 dt . \tag{35}$$

By the data processing inequality, a direct consequence of Lemma 3 is

**Corollary 2.**

$$\mathrm{KL}(\mu_T||\nu_T) \leq \mathrm{KL}(\mu_0||\nu_0) + \frac{1}{4\sigma}\mathbb{E}_{\mu_{[0,T]}}\int_0^T \|a(t, x) - b(t, x)\|^2 dt . \tag{36}$$

We first apply Corollary 2 from $t = 1$ to $t = 0$ to the two reverse Fokker-Planck equations (27) and (32), respectively sending $\mu$ back to $\pi$, and the current model sending $\mu$ back to $\pi^{(k+1)}$. Since they share the same initial condition, we have

$$\mathrm{KL}(\pi||\pi^{(k+1)}) \leq \frac{1}{4\epsilon}\int_0^1 \mathbb{E}_{\pi_t} \|f^*(t, x) - \hat{f}^{(k)}(t, x) - 2\epsilon(s^*(t, x) - \hat{s}^{(k)}(t, x))\|^2 dt . \tag{37}$$

We now apply Lemma 1 to the pair of forward Fokker-Planck equations (25) and (31), to obtain

$$\mathrm{KL}(\pi||\pi^{(k)}) = \mathrm{KL}(\mu||\mu^{(k)}) + \epsilon\,\mathbb{E}\int_0^1 \|\nabla\log\pi_t - \nabla\log\pi_t^{(k)}\|^2 dt \tag{38}$$

$$- \mathbb{E}_\pi\int_0^1 \langle f^* - f^{(k)}, \nabla\log\pi_t - \nabla\log\pi_t^{(k)}\rangle dt$$

$$= \mathrm{KL}(\mu||\mu^{(k)}) + \epsilon\,\mathbb{E}\int_0^1 \|s_t^* - s_t^{(k)}\|^2 dt$$

$$- \mathbb{E}_\pi\int_0^1 \langle f^* - f^{(k)}, s_t^* - s_t^{(k)}\rangle dt .$$

From (37) we have

$$\mathrm{KL}(\pi||\pi^{(k+1)}) \leq \frac{1}{4\epsilon}\|f^* - \hat{f}^{(k)}\|_\pi^2 + \epsilon\|s^* - \hat{s}^{(k)}\|_\pi^2 - \langle f^* - \hat{f}^{(k)}, s^* - \hat{s}^{(k)}\rangle_\pi . \tag{39}$$

Assuming a drift and score approximation error uniformly bounded by $\delta$, plugging (38) we have

$$\mathrm{KL}(\pi||\pi^{(k+1)}) \leq \frac{1}{4\epsilon}\mathbb{E}\|f^* - f^{(k)}\|^2 + \epsilon\|s^* - s^{(k)}\|^2 - \mathbb{E}\langle f^* - f^{(k)}, s^* - s^{(k)}\rangle \tag{40}$$

$$+ \delta^2\left(\frac{1}{4\epsilon} + \epsilon + 1\right) + \delta\left(\frac{1 + 2\epsilon}{2\epsilon}\|f^* - f^{(k)}\| + (1 + \epsilon)\|s^* - s^{(k)}\|\right) \tag{41}$$

$$\leq \mathrm{KL}(\pi||\pi^{(k)}) - \mathrm{KL}(\mu||\mu^{(k)}) \tag{42}$$

$$+ \frac{1}{4\epsilon}\mathbb{E}\|f^* - f^{(k)}\|^2 + C_1(\epsilon)\delta^2 + \delta(C_2(\epsilon)\|b^* - b^{(k)}\| + C_3(\epsilon)\|s^* - s^{(k)}\|) . \tag{43}$$

Now, using the condition number and SI Lipschitz assumptions, denoting $\eta = 1 + \frac{L}{4} - \chi^{-1}$, we obtain

$$\mathrm{KL}(\pi||\pi^{(k+1)}) \leq \eta \mathrm{KL}(\pi||\pi^{(k)}) + C_1(\epsilon)\delta^2 + 2\delta \tilde{C}_2(\epsilon)(\|b^* - b^{(k)}\| + \|s^* - s^{(k)}\|) \,, \tag{44}$$

with $C_1(\epsilon) = O(\epsilon + \epsilon^{-1})$ and $\tilde{C}_2(\epsilon) = O(1 + \epsilon^{-1})$

Observe that from (38) and using Cauchy-Schwartz, we have

$$\epsilon\|s^* - s^{(k)}\|^2 \leq (1 - \chi^{-1})\mathrm{KL}(\pi||\pi^{(k)}) + |\langle f^* - f^{(k)}, s^* - s^{(k)}\rangle| \tag{45}$$

$$\leq (1 - \chi^{-1})\mathrm{KL}(\pi||\pi^{(k)}) + \sqrt{L\epsilon \mathrm{KL}(\pi||\pi^{(k)})}\|s^* - s^{(k)}\| \tag{46}$$

$$\leq \eta \mathrm{KL}(\pi||\pi^{(k)}) + 2\sqrt{\eta\epsilon \mathrm{KL}(\pi||\pi^{(k)})}\|s^* - s^{(k)}\| \,, \tag{47}$$

which implies $\|s^* - s^{(k)}\| \leq 4\epsilon^{-1/2}\sqrt{\eta \mathrm{KL}(\pi||\pi^{(k)})}$, and therefore

$$\|b^* - b^{(k)}\| + \|s^* - s^{(k)}\| \leq C_3(\epsilon)\sqrt{\eta \mathrm{KL}(\pi||\pi^{(k)})} \tag{48}$$

with $C_3(\epsilon) = O(\epsilon^{1/2} + \epsilon^{-1/2})$. Thus, by redefining $\tilde{\delta} = \bar{C}_\epsilon\delta$ for $\bar{C}_\epsilon = O(\epsilon^{-3/2})$ we obtain

$$\mathrm{KL}(\pi||\pi^{(k+1)}) \leq \eta \mathrm{KL}(\pi||\pi^{(k)}) + \tilde{\delta}^2 + 2\tilde{\delta}\sqrt{\eta \mathrm{KL}(\pi||\pi^{(k)})} \tag{49}$$

$$= \left(\sqrt{\eta \mathrm{KL}(\pi||\pi^{(k)})} + \tilde{\delta}\right)^2 \,. \tag{50}$$

Setting $\alpha_k = \mathrm{KL}(\pi||\pi^{(k)})^{1/2}$, we arrive at the linear recurrence

$$\alpha_{k+1} \leq \sqrt{\eta}\alpha_k + \tilde{\delta} \,. \tag{51}$$

Solving this linear recurrence yields

$$\alpha_k \leq \eta^{k/2}\alpha_0 + \frac{\tilde{\delta}}{1 - \sqrt{\eta}} \,, \tag{52}$$

hence

$$\mathrm{KL}(\pi||\pi^{(k)}) \leq \left(\eta^{k/2}\alpha_0 + \frac{\tilde{\delta}}{1 - \sqrt{\eta}}\right)^2 \tag{53}$$

$$\leq 2\eta^k \mathrm{KL}(\pi||\pi^{(0)}) + \frac{2\tilde{\delta}^2}{(1 - \sqrt{\eta})^2} \tag{54}$$

$$= 2\eta^k \mathrm{KL}(\pi||\pi^{(0)}) + O(\epsilon^{-3})\frac{\delta^2}{(1 - \sqrt{\eta})^2} \,, \tag{55}$$

as claimed. ∎

*Proof of Lemma 1.* Let $K_t = \mathrm{KL}(\mu_t||\nu_t) = \int \mu_t(x) \log\left(\frac{\mu_t(x)}{\nu_t(x)}\right) dx$. By definition, the laws $\mu_t$ and $\nu_t$ solve the Fokker-Planck equations

$$\partial_t \mu_t = \nabla \cdot ((-b + \sigma\nabla\log\mu_t)\mu_t) \,, \tag{56}$$

$$\partial_t \nu_t = \nabla \cdot ((-a + \sigma\nabla\log\nu_t)\nu_t) \,. \tag{57}$$

We compute

$$\frac{d}{dt}K_t = -\int \frac{\mu_t(x)}{\nu_t(x)}\partial_t\nu_t(x)dx + \int \log\left(\frac{\mu_t(x)}{\nu_t(x)}\right)\partial_t\mu_t(x)dx \tag{58}$$

$$= -\int \frac{\mu_t}{\nu_t}\nabla \cdot ((-a + \sigma\nabla\log\nu_t)\nu_t)dx + \int \log\left(\frac{\mu_t}{\nu_t}\right)\nabla \cdot ((-b + \sigma\nabla\log\mu_t)\mu_t)dx \tag{59}$$

$$= \int \langle\nabla\left(\frac{\mu_t}{\nu_t}\right), -a + \sigma\nabla\log\nu_t\rangle\nu_t dx - \int \left\langle\nabla\log\left(\frac{\mu_t}{\nu_t}\right), (-b + \sigma\nabla\log\mu_t)\right\rangle\mu_t dx \tag{60}$$

$$= \int \left\langle\nabla\log\left(\frac{\mu_t}{\nu_t}\right), -a + b - \sigma\nabla\log\left(\frac{\mu_t}{\nu_t}\right)\right\rangle\mu_t \tag{61}$$

$$= -\sigma \mathrm{I}(\mu_t||\nu_t) + \mathbb{E}_{\mu_t}\langle b - a, \nabla\log\mu_t - \nabla\log\nu_t\rangle \,. \tag{62}$$

∎

# B CASE STUDY: AWGN WITH GAUSSIAN PRIOR

We illustrate our algorithm and the theoretical guarantees using arguably the simplest setting of a Gaussian prior $\pi = \mathcal{N}(b, \Sigma)$ and the AWGN channel. Since the only relevant quantity is the signal-to-noise ratio, we can assume without loss of generality that $\mathcal{F}(x) = x + w$, with $w \sim \mathcal{N}(0, I)$ (i.e., we renormalize the signal so that the noise variance is 1).

**Linear Convergence and ODE advantage**  We suppose that $\Sigma \succ 0$, and we consider a Gaussian initialization $\pi_0 = \mathcal{N}(b_0, \Sigma_0)$ with $\Sigma_0 \succ 0$, and use the simple linear interpolant

$$I_t = (1 - \sqrt{t})x + \sqrt{t}\,\mathcal{F}(x) = x + \sqrt{t}\,w \,.$$

In the AWGN channel, the transport equation between $\pi$ and $\mathcal{K}\pi$ is straightforward, given by the heat equation $\partial_t \pi_t = \frac{1}{2}\Delta \pi_t$ with $\pi_0 = \pi$: that is, for any Gaussian measure $\tilde{\pi} = \mathcal{N}(\tilde{b}, \tilde{\Sigma})$, the law $\tilde{\pi}_t$ of $I_t$ is explicitly given by $\tilde{\pi}_t = \mathcal{N}(\tilde{b}, \tilde{\Sigma} + tI)$.

For any diffusion coefficient $\epsilon \geq 0$, the reverse transport equation is then

$$\partial_t \rho_t = \frac{1}{2}\nabla \cdot ((-(1 + \epsilon)\nabla \log \rho_t)\rho_t) - \frac{1}{2}\epsilon \Delta \rho_t \,, \tag{63}$$

running backwards from the terminal condition $\rho_1 = \mathcal{N}(\tilde{b}, \tilde{\Sigma} + I)$. The corresponding reverse SDE is

$$dY_{\tilde{t}} = \frac{1 + \epsilon}{2}\nabla \log \tilde{\pi}_{1 - \tilde{t}}(Y_{\tilde{t}})d\tilde{t} + \sqrt{\epsilon}dW_{\tilde{t}} \,, \tag{64}$$

where now $\tilde{t} = 0$ corresponds to measurements and $\tilde{t} = 1$ to data. For a Gaussian measure $\tilde{\pi}$, the score $\nabla \log \tilde{\pi}_t$ is an affine field given by

$$\nabla \log \tilde{\pi}_t(x) = -(\tilde{\Sigma} + tI)^{-1}(x - \tilde{b}) \,. \tag{65}$$

As a result, the solution of the SDE (64) is an Ornstein-Ulhenbeck process that can be explicitly integrated. Denoting by

$$\Phi(t, s) = (\tilde{\Sigma} + (1 - t)I)^{(1+\epsilon)/2}(\tilde{\Sigma} + (1 - t)I)^{-(1+\epsilon)/2} \tag{66}$$

the solution map from time $s$ to time $t$, we have

$$Y_1 = \tilde{b} + \Phi(1, 0)(Y_0 - \tilde{b}) + \sqrt{\epsilon}\int_0^1 \Phi(1, s)dW_s \,, \tag{67}$$

and therefore that the Gaussian structure is preserved throughout the iterations. Recalling that $\pi_{k+1} = \text{Law}(Y_1)$, with $\tilde{b} = b_k$, $\tilde{\Sigma} = \Sigma_k$ above, given current parameters $b_k$, $\Sigma_k$, we verify from (67) that the new parameters become

$$b_{k+1} = b_k + B_k(b - b_k) \quad, \quad \Sigma_{k+1} = \Sigma_k + B_k(\Sigma - \Sigma_k)B_k \,, \text{ with} \tag{68}$$
$$B_k = (\Sigma_k(\Sigma_k + I)^{-1})^{(1+\epsilon)/2} \,.$$

We now establish that the means $b_k$ and covariances $\Sigma_k$ converge respectively to $b$ and $\Sigma$ at a linear rate.

**Proposition 3** (Gaussian AWGN Linear Convergence). *Let $\eta = \min(\lambda_{\min}(\Sigma), \lambda_{\min}(\Sigma_0))$ and assume $\eta > 0$. Then*

$$\|\Sigma - \Sigma_k\| \leq (1 - \eta^{1+\epsilon}(1 + \eta)^{-1-\epsilon})^k\|\Sigma - \Sigma_0\| \,, \tag{69}$$
$$\|b - b_k\|^2 \leq (1 - \eta^{1+\epsilon}(1 + \eta)^{-1-\epsilon})^k\|b - b_0\|^2. \tag{70}$$

*Proof.* Denote by $r_k = b - b_k$ and $R_k = \Sigma - \Sigma_k$. We have $b_{k+1} = (I - B_k)b_k$ and $R_{k+1} = R_k - B_k R_k B_k$. We verify [4] that for any symmetric matrix $X$ (not necessarily psd) and symmetrc matrix $B$ such that $0 \prec B \prec I$ we have $\|X - BXB\| \leq (1 - \lambda_{\min}(B)^2)\|X\|$. Therefore

$$\|R_{k+1}\| \leq (1 - \lambda_{\min}(B_k)^2)\|R_k\| \,. \tag{71}$$

---

[4]Indeed, split $X$ into $X = X_p + X_n$ where $X_p \succeq 0$ and $X_n \preceq 0$. Then we verify that $0 \preceq \bar{X}_p := X_p - BX_pB \preceq (1 - \lambda_{\min}(B)^2)X_p$ and $0 \succeq \bar{X}_n := X_n - BX_nB \succeq (1 - \lambda_{\min}(B)^2)X_n$. It follows that $X - BXB = \bar{X}_p + \bar{X}_n$ with $(1 - \lambda_{\min}(B)^2)\lambda_{\min}(X) \leq \lambda_{\min}(\bar{X}_n) \leq \lambda_{\min}(X - BXB) \leq \lambda_{\max}(X - BXB) \leq \lambda_{\max}(\bar{X}_p) \leq (1 - \lambda_{\min}(B)^2)\lambda_{\max}(X)$.

We claim that, for all $k$,

$$\lambda_{\min}(\Sigma_k) \geq \min\left(\lambda_{\min}(\Sigma), \lambda_{\min}(\Sigma_0)\right) = \eta . \tag{72}$$

Indeed, this follows from a simple induction argument: since $B_k$ and $\Sigma_k$ commute, we have that

$$\Sigma_{k+1} = (I - B_k^2)\Sigma_k + B_k \Sigma B_k \tag{73}$$

$$\succeq \lambda_{\min}(\Sigma_k)(I - B_k^2) + \lambda_{\min}(\Sigma)B_k^2 \tag{74}$$

$$\succeq \min(\lambda_{\min}(\Sigma_k), \lambda_{\min}(\Sigma))I , \tag{75}$$

As a result, we immediately obtain

$$\lambda_{\min}(B_k) = \left(\frac{\lambda_{\min}(\Sigma_k)}{1 + \lambda_{\min}(\Sigma_k)}\right)^{(1+\epsilon)/2} \geq \left(\frac{\eta}{1+\eta}\right)^{(1+\epsilon)/2} , \tag{76}$$

establishing (69). ∎

We thus verify that the iterative scheme converges at linear rate, with a factor $1 - \eta^{1+\epsilon}(1+\eta)^{-1-\epsilon}$. The setting that optimizes this upper bound is given by $\epsilon = 0$, which corresponds to the ODE setting. Interestingly, this relative advantage of the ODE is also observed in our numerical experiments. In particular, in the low SNR regime where $\eta \ll 1$, the convergence rate is of order $(1 - \eta)^k$. As may be expected, in the high SNR regime the convergence rate is much faster, since the effect of the channel is relatively minor. Additionally, and since in the finite-dimensional representation of Gaussian measures all metrics are equivalent, from (69) one obtains linear convergence both in Wasserstein and in KL divergence.

**Comparison with EM** Another interesting aspect of the Gaussian AWGN example is that we can compare the performance of our scheme to the actual EM algorithm of Rozet et al. (2024). As in our scheme, the Gaussian structure is preserved throughout the iterations, and moreover it corresponds to a mirror descent algorithm Kunstner et al. (2021).

Focusing on the centered Gaussian setting for simplicity, the EM algorithm updates a current Gaussian prior $\pi_k$ with covariance $\tilde{\Sigma}_k$ by first considering the posterior

$$p_k(x|y) = \frac{\gamma_{0,\tilde{\Sigma}_k}(x)\gamma_{x,I}(y)}{\gamma_{0,\tilde{\Sigma}_k+I}(y)} , \tag{77}$$

where $\gamma_{b,\Sigma}$ denotes the Gaussian density with mean $b$ and covariance $\Sigma$. This posterior is then used to update the prior via the 'M' step, defining the mixture

$$\mathrm{d}\pi_{k+1}(x) = \mathbb{E}_{y\sim\mu}[p_k(\mathrm{d}x|y)] = \gamma_{0,\tilde{\Sigma}_k}(x) \, \mathbb{E}_{y\sim\gamma_{0,\Sigma+I}}\left[\frac{\gamma_{x,I}(y)}{\gamma_{0,\tilde{\Sigma}_k+I}(y)}\right] \mathrm{d}x . \tag{78}$$

We can explicitly compute the RHS, since it is a Gaussian integral, resulting in $\pi_{k+1} = \gamma_{0,\tilde{\Sigma}_{k+1}}$, with precision matrix $\tilde{\Sigma}_{k+1}^{-1} = \tilde{\Sigma}_k^{-1} + I - ((\Sigma+I)^{-1} - (\tilde{\Sigma}_k+I)^{-1} + I)^{-1}$. As a result, the covariances $\tilde{\Sigma}_k$ evolve under EM according to

$$\tilde{\Sigma}_{k+1} = \tilde{\Sigma}_k + M_k(\Sigma - \tilde{\Sigma}_k)M_k , \text{ with } M_k = \tilde{\Sigma}_k(\tilde{\Sigma}_k + I)^{-1} . \tag{79}$$

Notice that this EM update is identical to the update (68) for the choice $\epsilon = 1$. Crucially, the asymptotic rate of this choice is *suboptimal* in our class, as it is dominated by the choices $\epsilon < 1$; in particular, in the low SNR regime $\eta \ll 1$, EM has a rate of $(1 - \eta^2)^k$, in contrast with the $(1 - \eta)^k$ of the ODE ($\epsilon = 0$), indicating that the ODE marginal transport provides some form of acceleration. Beyond the asymptotic rates, the EM algorithm enjoys a non-asymptotic rate of $O(1/k)$ thanks to its mirror-descent structure. By analogy with the acceleration phenomenon in convex optimization, one could expect an accelerated non-asymptotic rate of $O(1/k^2)$ for the ODE scheme: we verify that this indeed the case empirically; see Figure 6, but this non-asymptotic analysis is out of the present scope.

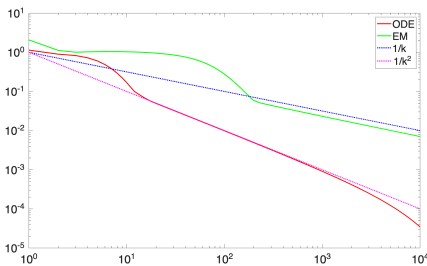

Figure 6: Convergence of $\|\Sigma - \Sigma_k\|^2$ using either ODE ($\epsilon = 0$, in red) or the EM algorithm ($\epsilon = 1$, green curve), for $\Sigma$ drawn from the Wishart distribution. In dashed we plot the rates $1/k^2$ and $1/k$ respectively.

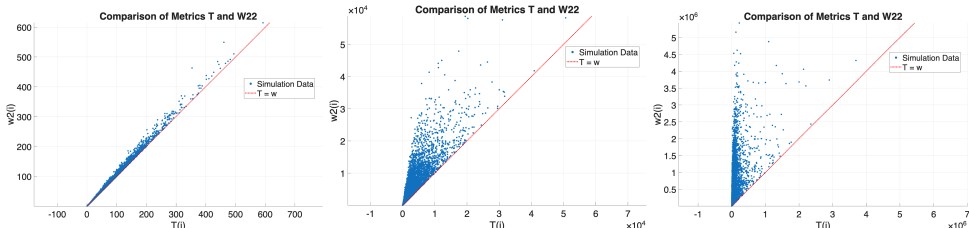

Figure 7: Scatterplot of $W_2^2(A, B)$ (y-axis) versus $\mathcal{T}(A; B)$ (x-axis), for pairs of independent Wishart matrices $A$ and $B$. Red line corresponds to $R = 1$. Left: Low SNR, Center: Moderate SNR, Right: High SNR. We observe that $R < 1$ with high probability.

**Verifying convergence assumptions**   Finally, the Gaussian AWGN example allows us to verify our assumptions from Section 4. Let us focus on the ODE setting, since it corresponds to the fastest regime. For a psd matrix $A$, we denote by $T_A = (A(A + I)^{-1})^{1/2}$ its associated solution map from (66). Recall that the Lipschitz Wasserstein assumption 1 compares, for a pair of distributions (here both assumed to be Gaussian) $\mathcal{N}(0, A)$ and $\mathcal{N}(0, B)$, the transport cost

$$\mathbb{E}_{y \sim \mathcal{N}(0, A+I)}[\|T_A(y) - T_B(y)\|^2] = \text{Tr}((T_A - T_B)(A + I)(T_A - T_B)) := \mathcal{T}(A; B)$$

to the squared-Wasserstein distance

$$W_2^2(\mathcal{N}(0, A), \mathcal{N}(0, B)) = \text{Tr}(A + B - 2(A^{1/2}BA^{1/2})^{1/2}) .$$

First, we notice that in the limit of the SNR going to zero, i.e., $\|A\|, \|B\| \to 0$, we have $\mathcal{T}(A; B) = \text{Tr}(A) + \text{Tr}(B) - 2\text{Tr}(A^{1/2}B^{1/2}) + O(\|A\|^2 + \|B\|^2)$, while $W_2^2 = \text{Tr}(A) + \text{Tr}(B) - 2\|A^{1/2}B^{1/2}\|_*$. From the variational characterization $\|Y\|_* = \sup_{\|X\| \leq 1} \langle Y, X \rangle \geq \langle Y, I \rangle = \text{Tr}(Y)$, we thus verify that $R$ cannot be uniformly upper bounded by 1 in the PSD cone. However, for macroscopic SNR, we simulate pairs $(A, B)$ from the Wishart ensemble and verify numerically that $\mathcal{T}(A; B) < W_2^2(A, B)$ with overwhelming probability, even for moderately small SNRs; see Figure 7.

Similarly, the condition number (18) needed in the KL guarantees can be explicitly bounded in this setting. Indeed, for $\pi = \gamma_\Sigma$ we have

$$\chi = \sup_{\gamma_{\tilde{\Sigma}}} \frac{\text{KL}(\pi||\gamma_{\tilde{\Sigma}})}{\text{KL}(\mathcal{K}\pi||\gamma_{I+\tilde{\Sigma}})} \leq (1 + \lambda_{\min}(\Sigma)^{-1})^2 \simeq \eta^{-2} . \tag{80}$$

By plugging the rate $(1 - \chi^{-1})^k$ from Corollary 1, corresponding to a Fokker-Plank channel with $\epsilon = 1$ [5], we thus recover the previous rate.

---

[5] note that in the Fokker-Plank channels, the diffusion coefficient $\epsilon$ is part of the channel, and cannot therefore be tuned for inference as in the general SI setting.

## C  DETAILED ALGORITHM DESCRIPTION

### C.1  ALGORITHM PSEUDOCODE

The following pseudocode summarizes the full training procedure for our SCSI model in settings where the observation space $\mathcal{Y}$ can be embedded into $\mathcal{X}$ without loss of information.

---

**Algorithm 2:** Training of Self-Consistent Stochastic Interpolant (SCSI)

---

**Input**    : Observation distribution $\mu$, Forward mapping $\mathcal{F}$, Interpolant schedule $(\alpha, \beta, \gamma)$,
              Initialization of drift and denoiser $\Theta^0 = \{b^{(0)}, g^{(0)}\}$, Total number of iterations $K$,
              Number of transport steps $T_{\mathrm{tr}}$
**Output** : Optimized networks $\Theta^{(K)} = \{b^{(K)}, g^{(K)}\}$

1  $\Theta \leftarrow \Theta^{(0)}$                                                                    // Initialize transport map
2  **for** $k$ *in* $1 \ldots K$ **do**
3      **for** $i$ *in* $1 \ldots T_{\mathrm{tr}}$ **do**
4          $y \sim \mu$
5          $x = \Phi_{\Theta^{(k-1)}}(y)$                            // Backward transport to get a data sample
6          $\tilde{y} = \mathcal{F}(x)$                                                  // Map back to observations
7          $z \sim \mathcal{N}(0, 1); \ t \sim \mathcal{U}(0, 1)$
8          $I_t = \alpha_t x + \beta_t \tilde{y} + \gamma_t z$
9          SGD update of $\Theta$ via losses (3)(4)
10      $\Theta^{(k)} \leftarrow \Theta$                                                    // Update transport map
11  **return** $\Theta^{(K)}$

---

### C.2  CONDITIONAL GENERATION VIA LIFTING

In this subsection, we clarify how to proceed when embedding $\mathcal{Y}$ into $\mathcal{X}$ is not straightforward. As noted at the beginning of Section 2.2, one can always work in a common state space through a lifting construction, which also makes explicit how the method naturally yields conditional generation. Throughout this subsection, we therefore do not assume $\mathcal{X} = \mathcal{Y}$.

We lift the problem to a shared product space by defining $\Omega := \mathcal{X} \times \mathcal{Y}$ and introducing the map $\tilde{\mathcal{F}}(x, y) := (w, \mathcal{F}(x))$, where $w$ is drawn independently from an arbitrary base measure $\pi_0 \in \mathcal{P}(\mathcal{X})$. Let $\tilde{\pi}$ denote the joint distribution of $(x, y)$, so that its $\mathcal{X}$-marginal $\pi$ is the target distribution we want to generate. Under the lifted observation channel $\tilde{\mathcal{F}}$, the observed distribution becomes $\tilde{\mu} = \pi_0 \otimes \mu$. To illustrate the resulting algorithm, consider the analogue of (7), in which we do have access to samples $\tilde{x} = (x, y) \sim \tilde{\pi}$. In this setting, we introduce a lifted interpolant $\tilde{I}_t = (I_t^{(1)}, I_t^{(2)})$ by

$$I_t^{(1)} = \alpha_t x + \beta_t w, \quad t \in [0, 1], \tag{81}$$

$$I_t^{(2)} \equiv y, \qquad\qquad t \in [0, 1]. \tag{82}$$

In the ODE setting, the velocity field satisfies $\tilde{b}(t, \tilde{x}) = \mathbb{E}[\dot{\tilde{I}}_t | \tilde{I}_t = \tilde{x}] = (b(t, x, y), 0)$. Operationally, this amounts to appending the observation $y$ as an additional input to the transport model with state space $\mathcal{X}$, as commonly done in conditional generation in score-based diffusion models Song et al. (2021) and stochastic interpolants Albergo et al. (2023b). When $\tilde{\pi}$ is not directly accessible, as in our inverse generative setting, the same iterative self-consistency scheme introduced in Section 3.1 can be applied to the lifted formulation without modification.

## D  IMPLEMENTATION DETAILS

### D.1  ARCHITECTURE OF MODELS

We give the architecture details of our SI and diffusion model here. Both architectures are the U-net from Dhariwal & Nichol (2021), specifically following the implementation here. The main difference is that we reduce the number of model channels in the first layer from default 192 to 96 for the diffusion model and 64 for the stochastic interpolant. This is primarily done for computational

reasons. As a result, the small model (64 channels) has ~32 million parameters while the large model has ~70 million parameters. Maximum positional embedding for the diffusion model and SI is taken to be 10,000 and 2 respectively.

For 2-D latent parameters as used in random masking, we process them with a small U-net consisting of 2 convolution blocks sandwiched between two mode convolution layers and the number of channels given by channel multiplier. We concatenate this with the image along channel dimension. For 1-D latents as used in motion blur and JPEG compression, we process them with a three layer perceptron and then add them to the time embedding.

Table 3: Model configuration parameters

| Parameter | Value |
| --- | --- |
| Model channels | 96 (64) |
| Channel multiplier | [1, 2, 3, 4] |
| Channel multiplier for embeddings | 4 |
| Number of blocks | 3 |
| Attention on resolutions | [32, 16, 8] |
| Dropout Fraction | 0.10 |
| Max positional embedding | 10000 (2) |
| Number of channels in latent U-Net | 8 |

### D.2 TRAINING PARAMETERS

We use the same hyperparameters for all the experiments. The backward transport map via ODE or SDE is performed in 64 steps. For experiments with ODE, we choose the schedule of SI as $\alpha_t = 1 - t, \beta_t = t, \gamma_t = 0$. For experiments with SDE, we keep the same schedule for $\alpha_t, \beta_t$, set $\gamma_t = t(1 - t)$, and use $\epsilon = 0.1$.

We experimented with different choices of $T_{\mathrm{tr}}$, the number of backward transport steps in Alg. 1, and observed only minor differences across values, with $T_{\mathrm{tr}} = 1$ already sufficient for all current experiments. To support this observation, we revisited the experiment in Section 5.1 (Fig. 2) and varied $T_{\mathrm{tr}}$ while keeping the product $T_{\mathrm{tr}} \cdot K$ fixed so that the overall computational budget remained unchanged. The Wasserstein distance between the final solved distribution and the true distribution is summarized in the Table 4 below (three independent runs). The results show that the performance is quite insensitive to $T_{\mathrm{tr}}$ unless it becomes so large that it forces too few outer iterations. We also did not observe notable differences in convergence speed. Given this robustness and to avoid unnecessary hyperparameter tuning, we use $T_{\mathrm{tr}} = 1$ throughout this work.

Table 4: Effect of $T_{\mathrm{tr}}$ on final Wasserstein distance (three independent runs) for the synthetic example in Section 5.1 (Fig. 2)

| $T_{\mathbf{tr}}$ | Wasserstein Distance (Mean $\pm$ Std) |
| --- | --- |
| 1 | $0.0491 \pm 0.0038$ |
| 10 | $0.0460 \pm 0.0038$ |
| 100 | $0.0476 \pm 0.0047$ |
| 1000 | $0.0593 \pm 0.0021$ |

When $\Theta^{(k)}$ is far from the optimal at the early stages of the outer iteration, the distribution of self-generated observations $\mathcal{K}_{\mathcal{F}}(\Phi_{\Theta^{(k)}})_{\#}\mu$ may differ significantly from $\mu$, and consequently slow down the convergence in practice. To mitigate this effect, we modify the interpolant (8) by replacing $\mathcal{F}(\Phi_{\Theta^{(k)}}(y))$ with a mixture: with probability $p$ (set to 0.9 in our experiments), we use the generated observation, and with probability $1 - p$, we use the original $y$. As long as $p > 0$, following the same argument in Prop. 1, we know the fixed point still gives us the desired optimal parameters $\Theta^*$.

Furthermore, to enhance computational efficiency, for every data mapped back with ODE integration, we (re)-sample the observations twice to generate two interpolated points. This amortizes the cost of ODE integration, which is the most expensive step in the training process. We fix the learning rate to be 0.0005 and use cosine schedule with warmup. Random masking, motion blur and JPEG experiments are trained for 50,000 iterations while other experiments are trained for 20,000 iterations.

# E ADDITIONAL RESULTS

## E.1 IMPACT OF NETWORK SIZE

We first show the results for training a generative diffusion model using our network architecture and choices of hyperparameters like learning rate, number of iterations etc. This is to provide a simple intuition for the modeling capacity of our overall configuration. We train a big model (with 96 channels) and a small diffusion model (with 64 channels) on clean CIFAR-10 data. The big model corresponds to the model used for training a generative model on the restored samples, as well as for DPS experiments. Small model corresponds to the SI used in our experiments. The FID for samples generated with these models after training is 5.16 and 6.64 respectively. Note that these numbers are close to the FID reported in Table 2 for generative model trained on restored samples of random masking. In Fig. 8 and 9, we show some randomly drawn samples from these models.

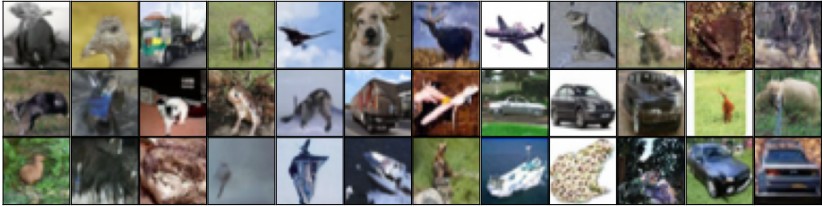

Figure 8: Randomly drawn images from the large diffusion model trained on cleaned images.

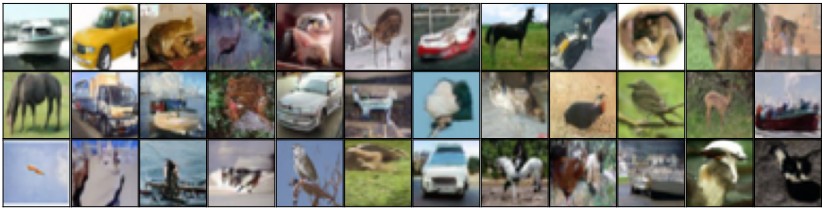

Figure 9: Randomly drawn images from the smaller diffusion model trained on cleaned images.

To see the impact of network capacity on our results, we repeat the random masking exercise with $\rho = 0.5$ (i.e., 50% pixels masked) with networks of different sizes. We vary the number of channels in the SI model from 48 to 96 which varies the number of network parameters in a 3x range from 17 million to 69 million. We train each network for 25k iterations and use ODE sampling.

Table 5 shows the FID between the restored samples and original CIFAR-10 results and we find that using a larger network improves the quality of restoration. Figure 10 shows the training loss curves for these and even though all the networks converge to similar loss, the loss for larger networks begins to drop earlier than for smaller networks. However note that here we focus on the number of iterations, and in terms of wallclock time, smaller networks do train much faster. Given these, in principle it is possible to design an iterative scheme which optimizes the wallclock time by training a smaller network and using its restored samples to warm-start a larger network.

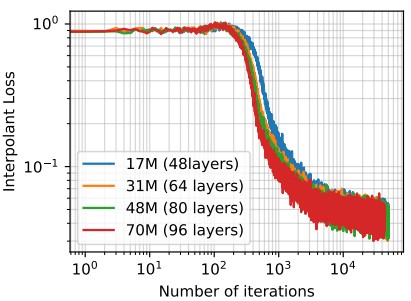

| Channels | FID |
|----------|------|
| 48 | 1.69 |
| 64 | 1.40 |
| 80 | 1.26 |
| 96 | 1.17 |

Table 5: FID for restored samples.          Figure 10: Training loss curves.

## E.2 SDE RESULTS

We present results for restoration using SDE in this section. We experiment with two different strategies: i) learning a combined drift for the velocity drift and score terms using the objective (5), and ii) learning different networks for drift and score separately using losses (3)(4). While the latter offers more flexibility in varying the noise schedule during inference, training two networks is at least twice as expensive, and we found no improvement over learning a single network for the combined drift. Hence we adopt the combined-drift formulation as our default for SDE.

In general, we find that, compared to ODE, SDE is more sensitive to the tuning of hyperparameters, specifically the noise schedule and number of transport steps. We have not thoroughly explored these choices for each experiment, and hence the results presented here are generally comparable or inferior to the ODE results presented in the main text. This merits more investigation in the future.

Results for CIFAR-10 and quasar spectra are shown in Figures 11 and 12, respectively. All results use the default choices of noise schedule $\gamma_t = \gamma_0 t(1 - t)$ with $\gamma_0 = 0.05$, $\epsilon_t = \gamma_t$, and $T_{\text{tr}} = 1$. For random mask and Gaussian blur, our approach is able to restore the images but the quantitative performance is worse than ODE (e.g., for Gaussian blur, SDE samples have FID of 16.8 as compared to 6.17 for ODE). However, for random motion, the restored images have visible artifacts. For quasar spectra, SDE-restored samples have MSE very close to ODE in the low-resolution/high-SNR setting, but noticeably higher MSE in the high-resolution/low-SNR setting.

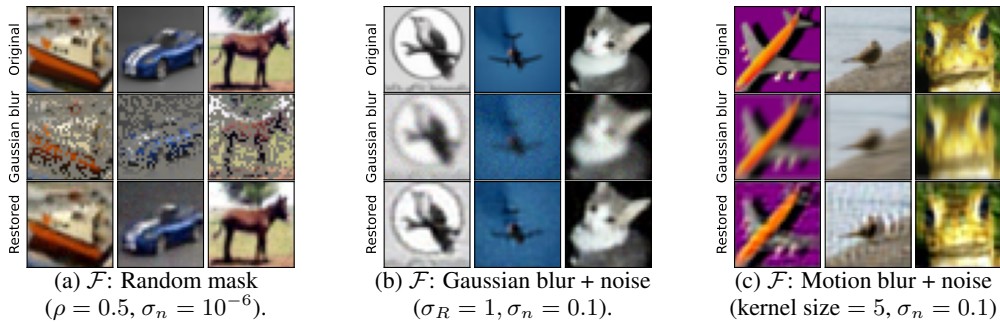

(a) $\mathcal{F}$: Random mask          (b) $\mathcal{F}$: Gaussian blur + noise          (c) $\mathcal{F}$: Motion blur + noise
($\rho = 0.5$, $\sigma_n = 10^{-6}$).          ($\sigma_R = 1$, $\sigma_n = 0.1$).          (kernel size = 5, $\sigma_n = 0.1$)

Figure 11: Restored samples for different forward maps with SDE based SI.

## E.3 ADDITIONAL METRICS FOR RESTORED PERFORMANCE COMPARISON

Table 6 reports additional quantitative metrics of imaging sample restoration for all corruption types, including comparisons with DPS and the SI-Oracle model. The SI-Oracle is trained using paired clean–corrupted data with the same SI architecture and serves as a performance upper bound for our method. We find that our method approaches this oracle for Gaussian blurring and random masking, indicating that the lack of clean supervision incurs only a modest penalty in these settings. The gap is more pronounced for random motion blur, suggesting this corruption poses greater challenges for unsupervised recovery.

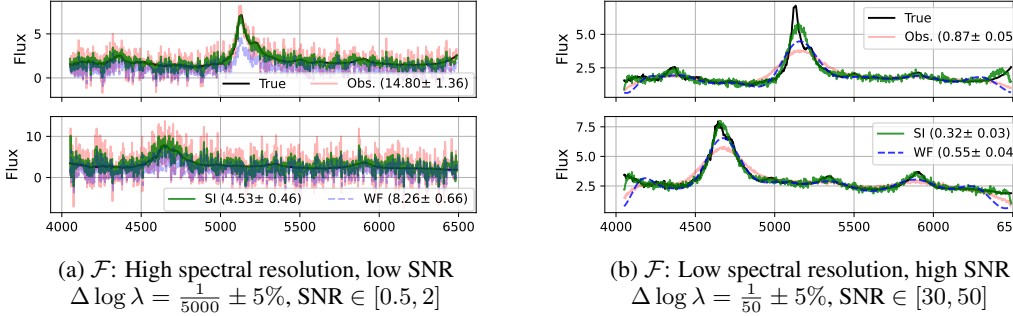

(a) $\mathcal{F}$: High spectral resolution, low SNR
$\Delta \log \lambda = \frac{1}{5000} \pm 5\%$, SNR $\in [0.5, 2]$

(b) $\mathcal{F}$: Low spectral resolution, high SNR
$\Delta \log \lambda = \frac{1}{50} \pm 5\%$, SNR $\in [30, 50]$

Figure 12: Restored quasar spectra using SDE. The numbers in the legend represent the mean squared error (MSE) averaged over 1000 true spectra.

Table 6: Quantitative comparison of individual-sample restoration quality across corruption types and noise levels with DPS and an oracle SI baseline.

| Forward Model | Metric | | Ours | DPS | SI-Oracle |
|---|---|---|---|---|---|
| Random masking ($\rho = 0.5, \sigma_n = 10^{-6}$) | LPIPS | ↓ | 0.00503 | 0.00497 | 0.00443 |
| | PSNR | ↑ | 28.97 | 29.4 | 29.4 |
| | SSIM | ↑ | 0.931 | 0.935 | 0.935 |
| | FID | ↓ | 1.57 | 1.47 | 0.85 |
| Random masking ($\rho = 0.5, \sigma_n = 0.1$) | LPIPS | ↓ | 0.00650 | 0.00776 | 0.00551 |
| | PSNR | ↑ | 27.8 | 26.7 | 28.4 |
| | SSIM | ↑ | 0.902 | 0.854 | 0.912 |
| | FID | ↓ | 2.78 | 17.54 | 1.07 |
| Gaussian blur ($\sigma_R = 1.0, \sigma_n = 0.1$) | LPIPS | ↓ | 0.00537 | 0.00949 | 0.00510 |
| | PSNR | ↑ | 29.2 | 26.8 | 28.8 |
| | SSIM | ↑ | 0.914 | 0.849 | 0.906 |
| | FID | ↓ | 6.74 | 29.9 | 1.33 |
| Gaussian blur ($\sigma_R = 1.0, \sigma_n = 0.25$) | LPIPS | ↓ | 0.0149 | 0.0249 | 0.0107 |
| | PSNR | ↑ | 26.5 | 24.1 | 26.1 |
| | SSIM | ↑ | 0.852 | 0.744 | 0.841 |
| | FID | ↓ | 14.5 | 60.52 | 1.57 |
| Motion blur ($k = 5, \sigma_n = 10^{-6}$) | LPIPS | ↓ | 0.00692 | 0.00275 | 0.00297 |
| | PSNR | ↑ | 28.9 | 33.9 | 32.1 |
| | SSIM | ↑ | 0.904 | 0.970 | 0.947 |
| | FID | ↓ | 7.7 | 0.98 | 2.47 |
| Motion blur ($k = 5, \sigma_n = 0.1$) | LPIPS | ↓ | 0.0108 | 0.0120 | 0.00297 |
| | PSNR | ↑ | 24.9 | 25.3 | 32.1 |
| | SSIM | ↑ | 0.804 | 0.796 | 0.947 |
| | FID | ↓ | 21.9 | 50.1 | 2.70 |

### E.4 SYNTHEIC MANIFOLD EXAMPLE

As noted in the main text, when the original forward model produces observations in a space different from that of the input variable, we may introduce an equivalent formulation in which both the input and output of the forward map lie in a common ambient space. To illustrate this procedure, we consider a synthetic example of Rozet et al. (2024), where the original forward model maps $\mathcal{X} = \mathbb{R}^5$ to $\mathcal{Y} = \mathbb{R}^2$. In this example, the ground truth distribution $\pi$ is supported on a one-dimensional curve in $\mathbb{R}^5$. Observations are generated as $y = Mx + \epsilon \in \mathbb{R}^2$, where $M \in \mathbb{R}^{2 \times 5}$ is a random matrix whose rows are sampled uniformly from $\mathbb{S}^4$, $\epsilon \sim \mathcal{N}(0, 10^{-4} I_2)$, and a different realization of $M$ is observed with each sample.

To construct a new observation in a five-dimensional space where our SCSI operates, we consider two simple embeddings: (1) *Direct padding*, where $y$ is augmented with three independent standard Gaussian coordinates; (2) *Adjoint embedding*, where $y$ is mapped to $M^\top y \in \mathbb{R}^5$. This choice still preserves all information in $y$ and is consistent with the general structure that quantities of the form $(\nabla_x \mathcal{F})^\top y$ always have the same dimension as $x$.

We follow the same setup in Rozet et al. (2024), using a dataset of 65536 independent observations, and train the transport model parameterized by a three-layer MLP of width 256. Figure 13 shows the reconstructed distributions through the 2D marginals $(x_1, x_2)$, $(x_2, x_3)$, and $(x_3, x_4)$. Both embeddings recover the underlying one-dimensional structure, with the adjoint embedding yielding a closer reconstruction in this setting.

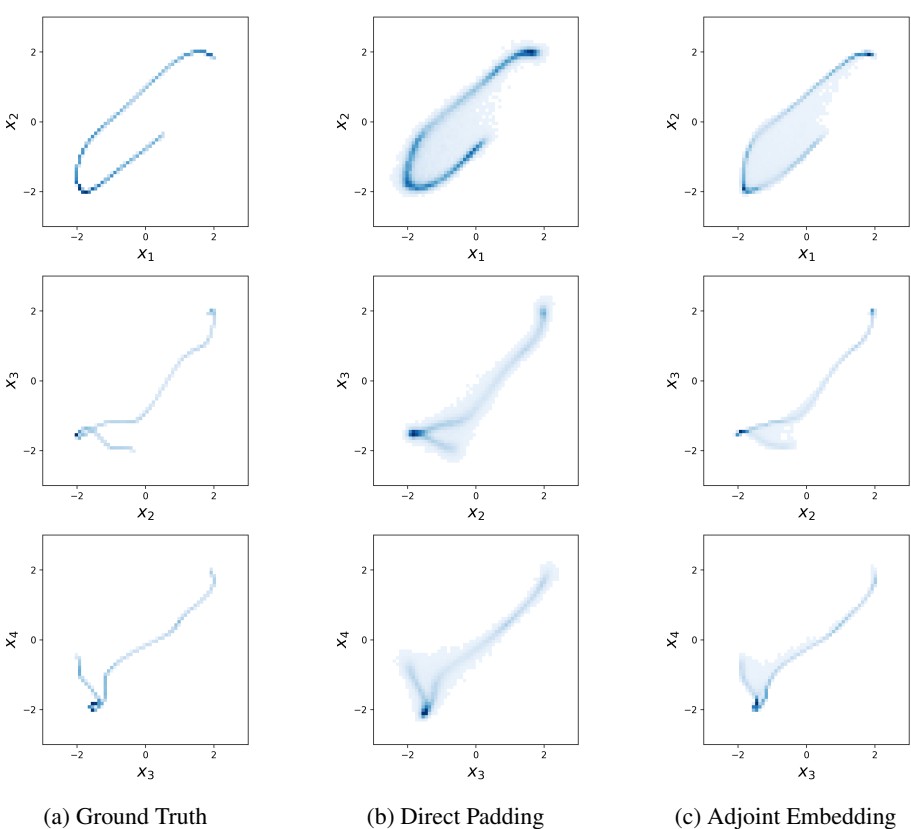

(a) Ground Truth      (b) Direct Padding      (c) Adjoint Embedding

Figure 13: Reconstruction on the synthetic manifold example of Rozet et al. (2024). Shown are three 2D marginals: $(x_1, x_2)$, $(x_2, x_3)$, and $(x_3, x_4)$ (from top to bottom) for the ground truth, the direct padding embedding, and the adjoint embedding. Both embeddings enable recovery of the one-dimensional curve, with the adjoint embedding producing the closer match.

## E.5   VARYING LEVELS OF CORRUPTION

In this section, we present additional results for varying the levels of corruption for different forward models used in the main-text.

### E.5.1   RANDOM MASKING

In Fig. 14, we show additional results for random masking experiment with 25%, 50% and 75% pixels randomly masked. The quality of restored images declines with increasing corruption, but the restored samples are close to original image even for 75% corruption.

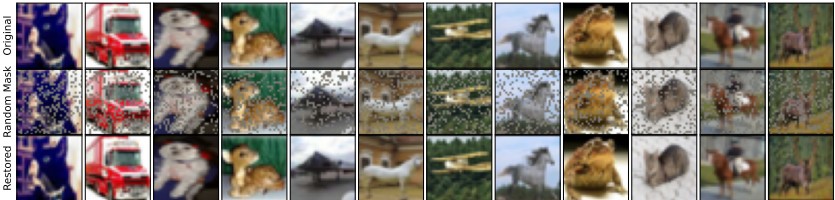

(a) Random masking with 25% pixels masked.

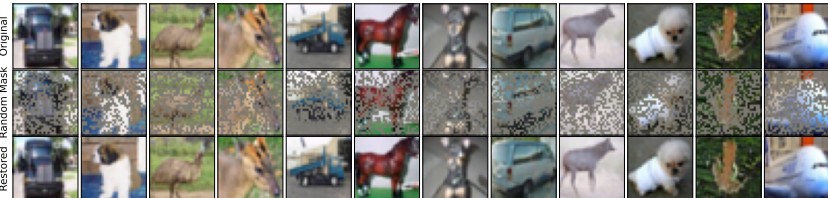

(b) Random masking with 50% pixels masked.

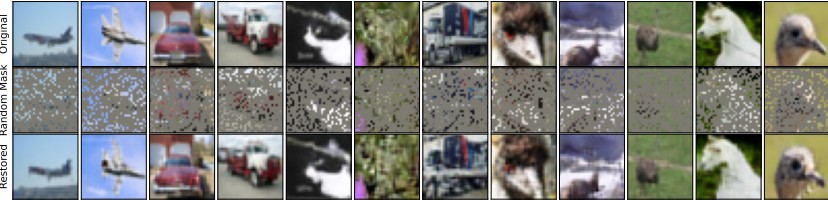

(c) Random masking with 75% pixels masked.

Figure 14: Restoring images with SI for varying fractions of masked pixels (levels of corruptions).

For generative modeling, we train a new diffusion model on the samples restored with SI. Fig 15 shows samples from the model trained on the restored samples of the random masking experiment with 50% corruption and negligible noise. As reported in the main text, FID of this model is 6.74.

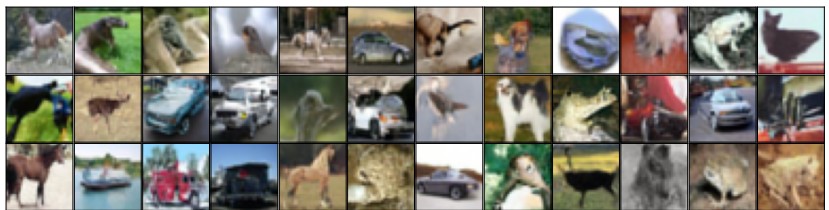

Figure 15: Samples from the diffusion model trained on the restored samples of random masking experiment with 50% corruption.

### E.5.2 MOTION BLUR

In Fig. 16, we show additional results for the motion blur experiment with increasing size of the motion blur kernel from 5 to 9 pixels.

### E.5.3 JPEG COMPRESSION

In Fig. 17, we show restorations for JPEG corruption for additional images that have been compressed with randomly chosen ratios. The SI is able to restore samples across a broad range of corruptions.

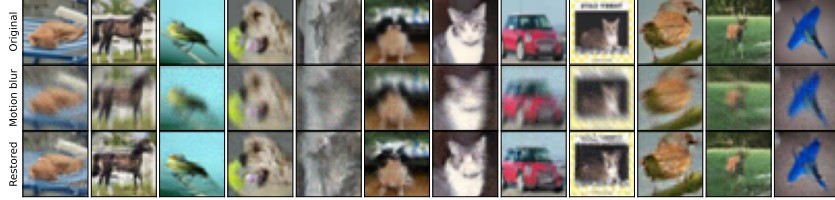

(a) Motion blur with blur kernel of 5 pixels.

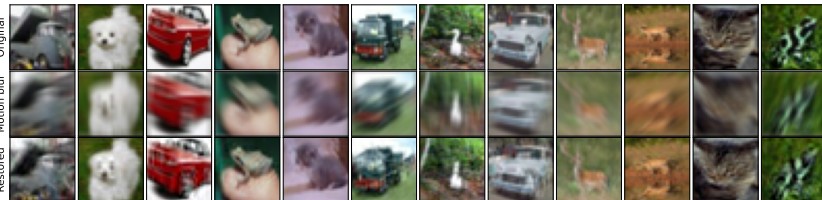

(b) Motion blur with blur kernel of 7 pixels.

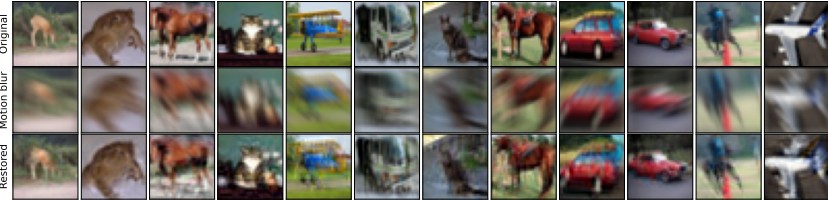

(c) Motion blur with blur kernel of 9 pixels.

Figure 16: Restoring images with SI for varying size of motion blur kernel (levels of corruptions).



Figure 17: Additional images for JPEG restoration for the model trained on samples with $q \sim \mathcal{U}[0.1, 1.]$.

In addition, we consider another setting where we have training samples that are corrupted with $q \sim \mathcal{U}[0.1, 0.5]$, i.e., we never see high quality samples. The results for the trained SI in this setting are shown in Fig. 18 and 19. The restoration for low-quality samples is poorer than when SI was trained on some samples with compression ratio of more than 0.5. However, note that the SI remains stable in the extrapolation range, i.e., when restoring sample of $q > 0.5$, the interpolant does indeed improve the restored image even though it has never seen samples in this regime.

### E.6 NON-GAUSSIAN NOISE: GAUSSIAN BLURRING WITH POISSON NOISE

In the main text, we considered only Gaussian additive noise to different forward models. In this section, we present additional results for when the forward map is blurring with a Gaussian kernel followed by adding Poisson noise. We add noise with two different levels, $\lambda_n = 0.1$ and $0.5$. The restored images here demonstrate that our approach also works in the non-Gaussian noise setting.

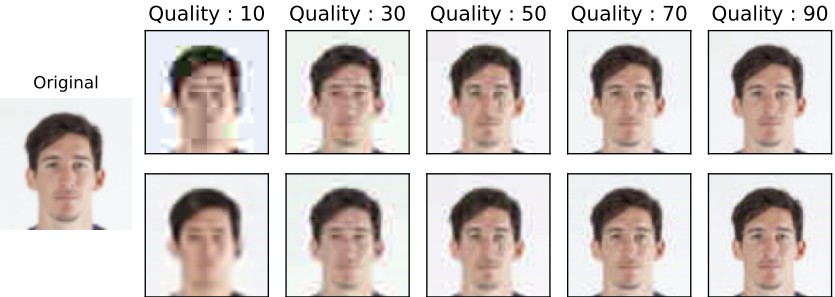

Figure 18: $\mathcal{F}$: JPEG compression + noise ($\sigma_n = 0.01$): results for different compression levels (Top: Corrupted; Bottom: Restored). Model is trained only on samples with $q \sim \mathcal{U}[0.1, 0.5]$. Results for higher qualities are in extrapolation regime.



Figure 19: Additional images for JPEG restoration for the model trained on samples with $q \sim \mathcal{U}[0.1, 0.5]$ only.

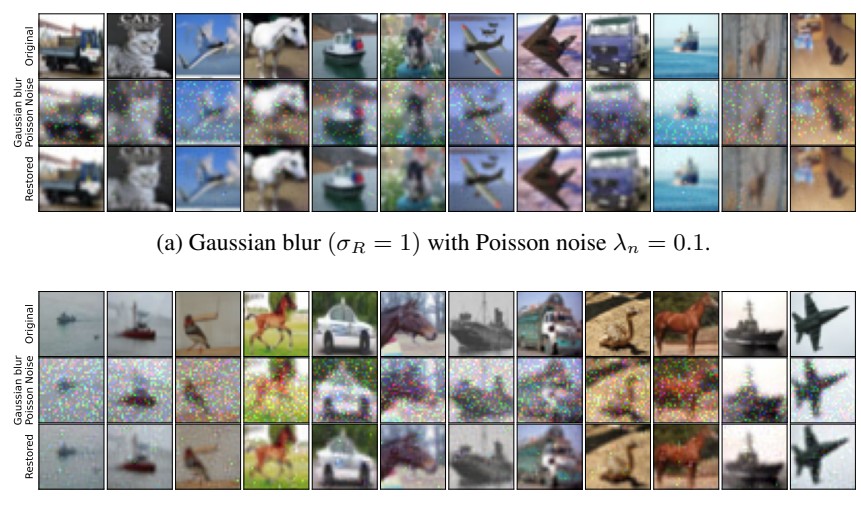

(a) Gaussian blur ($\sigma_R = 1$) with Poisson noise $\lambda_n = 0.1$.

(b) Gaussian blur ($\sigma_R = 1$) with Poisson noise $\lambda_n = 0.25$.

Figure 20: Restoring images with SI for Gaussian blurring with Poisson noise for different noise levels.

