# OpenReview forum: "Generative Modeling from Black-Box Corruptions via Self-Consistent Stochastic Interpolants"
_ICLR.cc/2026/Conference — ICLR 2026 Poster_

### Official Review · Reviewer_hPBy · 2025-10-29

**Soundness:** 3
**Presentation:** 3
**Contribution:** 3
**Rating:** 6
**Confidence:** 3

**Summary:**

This work addresses the problem of restoring clean samples from their corrupted measurements under two key constraints: 1) access is limited to the corrupted data, without corresponding clean samples, and 2) the forward map from the clean signal to the measurements is treated as avilable in a black-box manner. The authors approach this problem using Self-consistent Stochastic Interpolants, an extension of standard Stochastic Interpolants (SI) framework. This method adds a self-consistency mechanism, ensuring that the trained model's outputs map back to the true measurements. The method is introduced with rigorous mathematical formalism, solid theoretical results, and a broad analysis of convergence guarantees. The experimental section covers both standard benchmarks and a sophisticated application to quasar spectra recovery.

**Strengths:**

S1. The considered problem of obtaining clean samples when only the measurements and the black-box forward model are available seems important and challenging, yet the authors' approach based on self-consistency is both novel and effective.

S2. The proposed approach has strong theoretical foundations and is accompanied by a detailed convergence analysis.

S3. While the paper often utilizes more sophisticated mathematical tools, the authors took great care in writing the paper in a clear way with good flow.

**Weaknesses:**

W1. My primary concern relates to the lack of evaluation of the SDE version of the algorithm. The theoretical analysis focuses mainly on the SDE case, so it is natural to ask whether these results translate to practice. Moreover, as the authors also mention, the toy example (section 5.1) indicates the superiority of the SDE-based approach, and it seems counterintuitive that no other non-toy evaluations appeared in the paper. While I understand that it might be more computationally demanding and slower in terms of convergence, some kind of comparison in a standard experiment (section 5.2) should be provided.

W2. Equally concerning is the lack of standard evaluation metrics used in typical inverse-problems-related papers.

1. For the standard benchmarks (Table 1), only LPIPS is used. Methods for inverse problems are typically evaluated with the perception-distortion tradeoff in mind [1], so I would recommend adding PSNR, SSIM (distortion) and FID (perception).

2. For the comparison with other inverse generative models, only FID is provided. This is not crucial, but some metric like the Inception Score [2] should also be provided for a more balanced comparison.

3. For the quasar spectra recovery, no quantitative results are given. I'm not an expert in this subfield, but I expect the authors to provide proper metrics used in the evaluation scheme in such scenarios.

W3. The scope of the chosen baselines is very small. DPS is a good starting point, but as the authors correctly point out, its assumptions largely differ from their work. It would be beneficial for the paper to also include other methods that better cover the spectrum. Examples include: unsupervised methods that make different assumptions about the forward model [3,4], supervised bridge methods that ignore the forward model [5,6,7,8] or assume access to it [9], unpaired-data-based bridges [10,11] and the approaches explicitly cited by the authors as the closest ones in terms of assumptions and recency (lines 044-053). I don't expect the authors to include all of the above, but the comparison should be more representative.

W4. (Minor) I believe that the black-box assumption is lost when the method is additionally using either the mask in the random masking experiment or the compression magnitude in the JPEG one. The text should be slightly rephrased to emphasize that.

[1] Blau and Michaeli, The Perception-Distortion Tradeoff, CVPR, 2018

[2] Salimans et al., Improved Techniques for Training GANs, NeurIPS, 2016

[3] Wang et al., Zero-shot image restoration using denoising diffusion null-space model, ICLR, 2023

[4] Song et al., Pseudoinverse-guided diffusion models for inverse problems, ICLR, 2023

[5] Liu et al., I2SB: Image-to-image Schrodinger Bridge, ICML, 2023

[6] Zhou et al., Denoising Diffusion Bridge Models, ICLR, 2024

[7] Luo et al., Image restoration with mean-reverting stochastic differential equations, ICML, 2023

[8] Yue et al., Image restoration through generalized ornstein-uhlenbeck bridge, ICML, 2024

[9] Sobieski et al., System-embedded Diffusion Bridge Models, arXiv, 2025

[10] De Bortoli et al., Schrodinger bridge flow for unpaired data translation, NeurIPS 2024

[11] Kim et al., Unpaired image-to-image translation via neural schrodinger bridge, ICLR, 2024

**Questions:**

I think that the method is very elegant and I am generally sympathetic to the paper. However, I think some important points should be addressed, which I mentioned in the Weaknesses above. I have one additional question related to the theoretical results.

Q1. How do the considerations from lines 308-319 apply when the considered SDE has matrix-valued coefficients? Moreover, how do these results connect with the matrix-valued SDE proposed in [2], which embeds the Gaussian linear forward model into the coefficients?

[1] Song et al., Score-Based Generative Modeling through Stochastic Differential Equations, ICLR, 2021

[2] Sobieski et al., System-embedded Diffusion Bridge Models, arXiv, 2025

---

> ### Author Response · Authors · 2025-11-22
> **Response 1/2**
>
> We thank the reviewer for their careful and thorough review, for recognizing the strengths of our method, and for providing valuable suggestions. We hope the clarifications below help address the concerns raised, and we would appreciate your consideration in updating the score if they find the responses satisfactory.
>
> **Weakness 1.** Evaluation of SDE.
>
> **A.** We thank the reviewer for this important point, and we agree that SDE evaluations are a natural and valuable direction given our theoretical framework. We have been working on SDE implementations and will include results from two complementary approaches in the revised manuscript: (i) learning separate score and drift networks, and (ii) learning a combined effective drift that merges the velocity and score terms. Our preliminary findings suggest that the SDE formulation offers additional flexibility, though it also introduces new hyperparameter choices—particularly around noise schedules—that require careful tuning in high dimensions. Our current SDE results are slightly weaker than the ODE results, and we expect further improvements as we refine these choices.
>
> Interestingly, the strong performance of the ODE version is itself theoretically informative: it suggests that our guarantees may extend to the vanishing diffusion limit. As noted in the conclusions, formalizing this requires moving our analysis to the Wasserstein metric, which we view as a promising direction for future work.
>
> We will update the manuscript to include SDE comparisons and expand our discussion of the practical trade-offs between the two formulations.
>
> **Weakness 2.1.** More metrics in Table 1.
>
> **A.** We have re-run these experiments to evaluate PSNR and SSIM scores for Table 1. The conclusions remain unchanged and we find a similar trend to LPIPS in all the scores. We will add these results in the revision.
>
> **Weakness 2.2.** FID metric in Table 2.
>
> **A.** While we agree with the reviewer in principle, we have not reported other baselines as we have not run experiments on other inverse generative models, but instead used the FID results from respective papers directly. Since the original papers do not report other metrics, it is difficult to report these scores. Re-running these experiments ourselves would be very  expensive computationally, especially for the EM posterior setup (512 GPU hours as compared to our 86, without any gains to be made with parallelization given the sequential nature of the algorithm). That said, we are working on adapting the EM paper’s code to train the model from scratch ourselves and compute additional metrics for the masking setting, which is the only setting in which the EM method has been applied. We outperform Ambient diffusion by a large margin, so reporting different metrics there is less interesting.
>
> **Weakness 2.3.** Quantitative results for quasar spectra recovery.
>
> **A.** We have now included a Wiener filter restoration in our results, which is a standard baseline used for the current setup of the forward model. To quantify the gains, we report MSE of observed spectra, Wiener filtered spectra and our SI restored samples with the true spectra. These metrics for high-noise (and low-noise) setup are 11/6.5/0.65 (and 0.99/ 0.65/ 0.36) respectively, demonstrating the gains with our approach. We will update the figure in the revised paper with this baseline. In actual survey analysis, this restoration is done using complex posterior inference on object-by-object basis assuming a prior spectral distribution, which is beyond the scope of this work.

---

> ### Author Response · Authors · 2025-11-22
> **Response 2/2**
>
> **Weakness 3.** Other baselines.
>
> **A.** We thank the reviewer for this point. The fundamental distinction of our work is that we operate in a setting where clean samples are unavailable—we learn to restore from corrupted data alone. All the baselines mentioned by the reviewer, including DPS and the bridge-based methods [5–11], require access to clean samples, either for training (supervised bridges) or as the target distribution (unsupervised methods like DPS). This places them in a strictly easier regime and makes direct comparison somewhat apples-to-oranges: these methods solve a different problem with stronger supervision.
>
> That said, we agree that contextualizing our restoration quality is valuable. To this end, we have added a new baseline: a stochastic interpolant trained on paired (clean, corrupted) samples, similar in spirit to reference [5] but using our interpolant framework rather than a Schrödinger bridge. This represents an upper bound on what an interpolant-based method with comparable architecture and compute can achieve when clean data is available. The gap between our method and this oracle baseline quantifies the cost of operating without clean samples.
>
> The results are summarized in the table below, where SI-IGM denotes our method and SI-OPT denotes the “upper-bound’’ result described above. We find that our method approaches this oracle for Gaussian blurring and random masking, indicating that the lack of clean supervision incurs only a modest penalty in these settings. The gap is more pronounced for random motion blur, suggesting this corruption poses greater challenges for unsupervised recovery. We will include these comparisons and discussion in the revised manuscript.
>
> **Restored Performance Comparison**
>
> | Experiment | Metric | DPS | SI-IGM | SI-OPT |
> |-----------|--------|------|--------|---------|
> | **Random masking (low noise)** | LPIPS ↓ | 0.00497 | 0.00503 | 0.00443 |
> | | PSNR ↑ | 29.4 | 28.97 | 29.4 |
> | | SSIM ↑ | 0.935 | 0.931 | 0.935 |
> | **Random masking (high noise)** | LPIPS ↓ | 0.00776 | 0.00650 | 0.00551 |
> | | PSNR ↑ | 26.7 | 27.8 | 28.4 |
> | | SSIM ↑ | 0.854 | 0.902 | 0.912 |
> | **Gaussian blur (low noise)** | LPIPS ↓ | 0.00949 | 0.00537 | 0.00510 |
> | | PSNR ↑ | 26.8 | 29.2 | 28.8 |
> | | SSIM ↑ | 0.849 | 0.914 | 0.906 |
> | **Gaussian blur (high noise)** | LPIPS ↓ | 0.0249 | 0.0149 | 0.0107 |
> | | PSNR ↑ | 24.1 | 26.5 | 26.1 |
> | | SSIM ↑ | 0.744 | 0.852 | 0.841 |
> | **Motion blur (low noise)** | LPIPS ↓ | 0.00275 | 0.00692 | 0.00297 |
> | | PSNR ↑ | 33.9 | 28.9 | 32.1 |
> | | SSIM ↑ | 0.970 | 0.904 | 0.947 |
> | **Motion blur (high noise)** | LPIPS ↓ | 0.0120 | 0.0108 | 0.00297 |
> | | PSNR ↑ | 25.3 | 24.9 | 32.1 |
> | | SSIM ↑ | 0.796 | 0.804 | 0.947 |
>
>
> **Weakness 4.** Black-box assumption lost with latent variables present.
>
> **A.**  We thank the reviewer for highlighting this, and will clarify it further. By “black box setting”, we mean that we only have access to inputs and outputs of a simulator, but do not have access to the functional/parametric form or gradients of the forward map. All our examples fall in this category and in that vein, the random mask and the quality factor of JPEG experiment are considered outputs of the channel. We will clarify this further in the revised version of the paper.
>
> **Question 1.** SDE with matrix-valued coefficients
>
> **A.** This is a very interesting question, thanks for bringing it up. Our considerations on the Fokker-Planck channel certainly extend to the matrix-valued setting. More specifically, the current statement already covers the case where the drift is an arbitrary non-linear mapping, and our analysis carries over to SDEs with matrix-valued, time-dependent diffusion coefficients without substantive changes. The main notable difference is that with constant scalar diffusion, the Girsanov and KL bounds simplify to L2-type path integrals because the diffusion term factors out of the time integrals.
>
> The paper you cited (which we were unaware of) is a nice example of how one can embed certain linear inverse problems (a specific instance of our ‘corruption operators’ ) into the language of transport-based models using the Fokker-Planck channel formalism (although their problem setup differs from ours; they do not assume only access to corrupted data). We will add this example in the revision.
>
> Finally, it is important to emphasize that the representation in terms of a Fokker-Plank defines a model for the channel. In that sense, one should view this section as an instance of how a model for the channel can be seamlessly integrated into our method, with improved guarantees relative to the agnostic/black box SI.

---

> > ### Comment · Reviewer_hPBy · 2025-11-23
> >
> > Thank you for answering my questions. I will wait with my final decision until the manuscript is updated with the mentioned additions.

---

> > > ### Author Response · Authors · 2025-11-26
> > >
> > > Dear Reviewer,
> > >
> > > Thanks again for your time and for the valuable feedback. We have updated the manuscript to address all the comments from you and the other reviewers. A summary of the changes, together with the newly obtained results from the long-running experiment, is included in the first official comment at the top. We hope the revisions and new findings provide a clearer basis for your updated evaluation, and we would be happy to discuss any further questions.

---

> > > > ### Comment · Reviewer_hPBy · 2025-11-27
> > > >
> > > > Dear Authors,
> > > > thank you for updating the manuscript. I need to follow up with two additional questions:
> > > >
> > > > 1. Are the quantitative results of the SDE version for the 5.2 tasks included somewhere? Right now, I am only able to localize the numerical values for the quasar spectra experiment.
> > > > 2. I believe Table 6 should include all of the metrics (FID is missing) to correctly present the whole picture. Is this doable?

---

> > > > > ### Author Response · Authors · 2025-12-03
> > > > >
> > > > > **Re: Evaluation of SDE.** The reviewer is correct that Appendix D.2 lacks quantitative metrics for SDE. To address this, we conducted a hyperparameter study using small networks (17M parameters) at 10,000 iterations, varying $\gamma_0$, transport steps, integrator choice, averaging strategies, and network architecture. The results consistently show ODE outperforming SDE to some extent (FID; lower is better):
> > > > > - Random masking (low noise): ODE 4.11 vs. SDE 6.04
> > > > > - Gaussian blur (low noise): ODE 11.4 vs. SDE 15.6
> > > > > - Random motion (low noise): ODE 15.86 vs. SDE 19.96
> > > > >
> > > > > These gaps likely reflect the additional complexity of tuning SDE hyperparameters. We will include extended ablation studies in the revision.
> > > > > Importantly, the ODE setup demonstrates strong practical performance across all high-dimensional experiments. Moreover, we have since obtained convergence guarantees for the ODE case using the Wasserstein distance (rather than KL divergence used for SDE), and it is now presented in Sec 4.1 of the updated manuscript. This combination of empirical robustness and theoretical guarantees establishes ODE as the recommended approach.
> > > > >
> > > > > **Re: FID for Table 6.** We have added these results to the table. The trends align with other metrics: our approach performs comparably to or better than DPS without requiring clean samples, and approaches oracle SI performance in several settings.

---

### Official Review · Reviewer_vQVs · 2025-10-29

**Soundness:** 3
**Presentation:** 3
**Contribution:** 3
**Rating:** 8
**Confidence:** 4

**Summary:**

The authors consider a problem of inverse generative modelling in which they have access to samples of the distribution of corrupted samples and access to a black-box corruption map. The authors propose and theoretically justify a novel iterative algorithm based on iterative learning and inference of a stochastic interpolant model until it generates data such that the corruption map produces exactly the distribution of corrupted samples. The authors further evaluate their approach on image setups with different corruption maps and one scientific setup.

**Strengths:**

- The proposed approach is novel and theoretically justified.
- The proposed approach does not require knowing or using the specific functional form of the corruption map used.
- The authors evaluate their method on a wide range of image setups and one scientific setup, showing that the proposed method solve the problem.

**Weaknesses:**

- All the setups are more like synthetic, rather than real setups mentioned in the introduction: “... medical imaging, where we observe tomographic projections of internal structures, astronomical observations affected by atmospheric distortion, and other measurement processes that introduce noise and information loss”. The quasar spectra setup also assumes the usage of the synthetically provided process F. While the considered setups clearly show that the method works, the addition of one in the mentioned real-life examples would further strengthen the method. Specifically, they would show that the requirements of the method (like injectivity of F) could be satisfied in real-life problems and that there is a direct practical result.

**Questions:**

Do you have any understanding of it it possible to extend this framework to discrete domains by considering, e.g., masking or uniform diffusions?

---

> ### Author Response · Authors · 2025-11-22
>
> We thank you for your careful and thorough review, for recognizing the strengths of our method, and for providing valuable suggestions. We hope the clarifications below help address the concerns raised in your review, and we would appreciate your consideration in updating the score if you find the responses satisfactory.
>
> **Weakness 1.** Real-life examples.
>
> **A.** Thank you for your question. Indeed, we have designed the experimental setup to try to reflect the broad applicability of our method to the best of our (computational) abilities. It is important to emphasize that our approach substantially broadens the scope of prior work, since we only require access to a simulator of the channel, rather than its model. In other words, we do not need to know the likelihood function $p(y | x)$, where $y = \mathcal{F}(x)$; we only require samples from $p(y|x)$. This also allows us to easily apply our approach to non-linear forward model. The correct analogy is *Simulation-Based-Inference$, which has substantially extended the scope of statistical inference in physical applications thanks to this flexibility.
>
> Regarding the quasar spectra example, the actual forward-model pipeline involves many components, and we selected the two corruptions with the largest contributions (telescope point-spread smoothing and additive noise). Additional corruptions include non-Gaussian noise, pixel saturation/masking, and amplitude offsets. We did not include these for simplicity, and incorporating them would also make it difficult to establish a meaningful baseline. Instead, for the current smoothing+noise forward model, we have added a Wiener-filter restoration as a standard baseline. To quantify the gains, we report MSE of observed spectra, Wiener filtered spectra and our SI restored samples with the true spectra. These metrics for high-noise (and low-noise) setup are 11/6.5/0.65 (and 0.99/ 0.65/ 0.36) respectively, demonstrating the gains with our approach. We will update the figure in the revised paper with this baseline.
>
>
> Finally, we note that the required injectivity is defined at the level of the induced operator on *probability measures*, not at the level of the sample-wise corruption map. Even when the map $x \mapsto \mathcal{F}(x) $ is non-injective (e.g., a projection losing dimensions), the operator $\mathcal{K}$ acting on distributions can still be injective, meaning that $\mathcal{K}\pi_1 = \mathcal{K}\pi_2 $​ implies $\pi_1 = \pi_2 $. In essence, an injective corruption operator ensures that the input data distribution can be uniquely recovered from the output data distribution. Many real-world problems satisfy this condition, including the the AWGN channel, the 'random projection' channel, and the 'erasure' channel; see our reply to Q1 of the reviewer RjDz for additional discussion.
>
>
> **Question 1.** Extension to discrete domains.
>
> **A.** This is a very good point. Indeed, our framework builds on top of the formalism of Stochastic Interpolants / Flow matching: as long as one can define a stochastic interpolation between the input domain and the output domain, our method applies. As the reviewer points out, there are several recent works that extend SIs to discrete domains, see eg [R1, R2], so our method can also be applied to corrupted discrete data given a forward simulator. We will include a comment about this in the conclusion section.
>
> [R1] Discrete Flow Matching, by Gat et al  https://arxiv.org/abs/2407.15595
>
> [R2] Any-order flexible length masked diffusion, by Kim et al https://arxiv.org/abs/2509.01025

---

> > ### Comment · Reviewer_vQVs · 2025-11-26
> >
> > Thank you for your response. I keep my high score for the paper.

---

### Official Review · Reviewer_L5cg · 2025-10-30

**Soundness:** 1
**Presentation:** 2
**Contribution:** 2
**Rating:** 0
**Confidence:** 5

**Summary:**

The paper addresses the problem of inferring a probability density function from data that is obtained by a forward mapping F.
The topic is interesting and important. But the paper contains serious mistakes and therefore cannot be published.

**Strengths:**

The main strength of the paper is in problem definition and casting. The rest has some serious errors

**Weaknesses:**

The main problem with this paper is that it has a significant error that propagates through the paper.
They define the path
I = \alpha x_t + \beta F(x) + \gamma z

But this path is wrong. In particular, F(x) is not even in the same space (functional space) of x and in the finite dim case often has a very different dimensions. Take the example you propose, tomography. F(x) is projection data and x is the image. The size of F(x) depends on the number of projection I take. This path does not make any sense!

Hint for next time - try the path I = \alpha x_t + \beta (\grad F)^T F(x) + \gamma z
This additional operation is crucial both in the finite and infinite setting

**Questions:**

see above

---

> ### Author Response · Authors · 2025-11-21
> **Explaining why the dimensional concern is unfounded**
>
> We thank Reviewer L5cg for their feedback. However, we must respectfully point out that the central criticism raised—namely, that our path definition is fundamentally flawed because $\mathcal{F}(x)$ and $x$ live in different spaces—reflects a misreading of our paper. The score of 0 (strong reject) with confidence 5 (absolutely certain) is therefore not only disproportionate to the actual content of our work, but also at odds with the assessments of the three other reviewers who assigned scores of 6, 8, and 8. We believe a more careful reading of problem setup would have avoided this misunderstanding.
>
> **Regarding the dimensional concern:** We explicitly address this issue in our problem setup (lines 73-79 of the original submission). We state:
>
> > "Since $\mathcal{F}$ is a channel that does not introduce additional information about $x$, we assume that the observation space $\tilde{\Omega}$ can be embedded back into the data space $\Omega$ in a way that preserves all information contained in $y$. That is, the embedding itself does not introduce any additional information loss beyond what is already incurred by $\mathcal{F}$. With a slight abuse of notation, we therefore redefine $\mathcal{F}$ as a map $\mathcal{F}: \Omega \to \Omega$."
>
> This embedding is standard practice in inverse problems and allows us to work with a consistent space throughout. For the tomography example the reviewer mentions, this embedding is straightforward: after obtaining projection data, we embed it back into the image space (e.g., via zero-padding in the Fourier domain or other appropriate embeddings that preserve the information content). The path $I_t= \alpha_t x + \beta_t \mathcal{F}(x) + \gamma_t z​$ is therefore well-defined in $\Omega $ as written.
>
> **Another resolution (independent of embedding):** Even without choosing a specific embedding, one can always reformulate the inverse problem in a common ambient space. Let $\pi\in\mathcal{P}(\mathcal{X})$ be the distribution of interest, and let $\mathcal{F}:\mathcal{X}\to\mathcal{Y}$ be any (possibly stochastic) forward channel. Consider the product space $\bar{\Omega}:=\mathcal{X}\otimes\mathcal{Y}$ and define a lifted channel $\tilde{\mathcal{F}}(x,y):=(w,\mathcal{F}(x))$,
> where $w$ is drawn from an arbitrary base measure $\rho_0\in\mathcal{P}(\mathcal{X})$, independent of $\mathcal{F}(x)$. If $\mathcal{K}, \tilde{\mathcal{K}}$ denote the integral operator associated with $\mathcal{F}$ and $\tilde{\mathcal{F}}$ respectively, and $\nu\in\mathcal{P}(\bar{\Omega})$, then $\tilde{\mathcal{K}}\nu=\rho_0\otimes \mathcal{K}\nu_{\mathcal{X}}$, where $\nu_{\mathcal{X}}$ is the $\mathcal{X}$-marginal. Thus any solution $\tilde{\pi}$ of $\tilde{\mathcal{K}}\tilde{\pi}=\tilde{\mu}$ where $\tilde{\mu}=\rho_0\otimes\mu$, necessarily satisfies $\mathcal{K}\tilde{\pi}_{\mathcal{X}}=\mu$. In other words, solving the lifted inverse problem on $\bar{\Omega}$ is equivalent to solving the original inverse problem on $\mathcal{X}$, and the $\mathcal{X}$-marginal of $\tilde{\pi}$ recovers the desired solution. This shows that one can always work in a common space without altering the inverse problem. Our path definition is therefore mathematically consistent even without choosing an explicit embedding.
>
> **Regarding the reviewer's suggested alternative:** The reviewer proposes using $I_t= \alpha_t x + \beta_t [\nabla \mathcal{F}(x)^T] \mathcal{F}(x) + \gamma_t z​$. While this is an interesting suggestion, implementing it in our framework would require replacing the unknown $x$ with our denoiser $X_0(y)$, yielding:
> $$
> I_t = \alpha_t X_0(y) + \beta_t [\nabla_x \mathcal{F}(X_0(y))]^T y + \gamma_t z.
> $$
> This formulation, while potentially valuable, is significantly more complex to implement as it requires computing the gradient of the random channel evaluated on the denoised map.We appreciate this suggestion and are exploring simpler approaches for cases where an explicit embedding is not preferred.
>
> **In summary:** The reviewer's central objection is based on an oversight of our explicit discussion of space embedding, and the product-space construction above shows formally that the inverse generative modeling problem can always be posed in a single space without changing its meaning. We would welcome further technical dialogue, but the current review does not provide grounds for a score of 0. We hope the reviewer will reconsider their assessment in light of the material actually presented in our paper.

---

> > ### Comment · Reviewer_hPBy · 2025-11-21
> > **Poor review standard**
> >
> > I would like to voice my support for the authors' rebuttal regarding Reviewer L5cg's comments. I believe the review in question falls short of the constructive standards expected at ICLR for several reasons:
> >
> > 1. Lack of Engagement: The review does not appear to engage with the substance of the paper, focusing instead on arguments that lack technical depth.
> >
> > 2. Unsubstantiated Claims: The reviewer asserts that there are multiple 'serious mistakes' and 'errors' (using the plural form) but only identifies a single alleged issue—one which the authors have effectively refuted.
> >
> > 3. Unjustified Demands: The reviewer suggests modifications without providing a justification or technical reasoning for why those changes are necessary.
> >
> > 4. Score Discrepancy: A lowest-possible score with high confidence seems disproportionate, particularly given the evidence that the reviewer may have overlooked key sections of the paper.
> >
> > I encourage the Area Chair to weigh this review accordingly.

---

> > ### Comment · Reviewer_L5cg · 2025-11-27
> > **Strongly disagree**
> >
> > The idea that you can write a path \alpha x_t + \beta F(x) + \gamma z is **fundamentally  incorrect** and it is far from being a "standard" in inverse problems (you did not provide a reference for that). Your idea about embedding is not clear as well.
> > Zero padding is far from being the point and most of the time it is impossible. What is "an appropriate embeddings that preserve the information content" is unclear and needless to say that you did not introduced it in the paper.
> >
> > Take for example the seismic 3D FWI. In this case the data is obtained by the solution of the wave equation for different source and receiver configuration. The data is a tensor of dimension Sx \times Sy \times Rx \times Ry \times T where (Sx, Sy) are source configuration (Rx, Ry) are receiver configuration, T is the number of times recorded. The earth is a 3D tensor. How do you add F(x) which is a 5D tensor to x which is a 3D tensor?
> >
> > Similar problem occur in EIT and in DC resistivity. For those problems (for 2D solutions) the data, F(x), are 4D tensors (sx1, sx2, rx1, rx2) while x is a 2D tensor. In tomography, that we just discussed the number of projections can be larger or smaller than the number of pixels in the image. If it is larger than you have more data than unknowns.
> >
> > This is not semantics, F(x) can be added to x only for some trivial problems such as denoising or debarring. It is a **fundamental error in the paper**. Furthermore, if you look at most AI research in inverse problems (look for example at Deep Learning Techniques for Inverse Problems in Imaging, and the early work Solving ill-posed inverse problems using iterative deep neural networks) you will find that one of the main questions they ask is exactly that, how should we bring F(x) to the space of x so we can process it.
> >
> > I therefore stay with my score.

---

> > > ### Author Response · Authors · 2025-11-27
> > >
> > > We respectfully note that the embedding construction the reviewer claims is missing was explicitly stated in lines 73–79 of our original submission and expanded in our revision. The reviewer's continued assertion that we "did not introduce it in the paper" is factually incorrect.
> > >
> > >
> > > **Addressing the reviewer's examples directly:** The reviewer asks: "How do you add F(x), which is a 5D tensor, to x, which is a 3D tensor?" The answer is: we do not add them. Our product construction places them in a common space without requiring any such addition. Specifically, we define $\Omega := \mathcal{X} \times \mathcal{Y}$, so that both domains embed trivially into $\Omega$. The lifted channel is $\tilde{\mathcal{F}}(x, y) = (w, \mathcal{F}(x))$, where $w$ is drawn independently from a base measure on $\mathcal{X}$. This handles all the reviewer's examples—3D FWI, EIT, DC resistivity, and tomography with arbitrary projection counts—without exception and without requiring any `adding’ of mismatched objects.
> > >
> > >
> > > For the simpler case where both spaces are Euclidean with different dimensions, standard zero-padding (embedding $\mathbb{R}^k$ into $\mathbb{R}^d$ via $x \mapsto (x, 0)$) suffices. But even when no natural inclusion exists, the product construction above is fully general.
> > >
> > >
> > > **On the claim that this is "far from standard:"** Product-space constructions are routine in modern generative modeling. Diffusion models augment data with noise levels; text-to-image models jointly embed text tokens and image patches into a common latent space; multimodal architectures (e.g., for cryo-EM) combine GNN and image branches operating on heterogeneous domains. The idea that source and target must share a single ambient dimension to define a valid interpolant path reflects a misunderstanding of our framework.
> > >
> > >
> > > **In summary:** The product construction $\Omega = \mathcal{X} \times \mathcal{Y}$ handles all the reviewer's examples. This is not a workaround—it is the natural and fully general formulation, explicitly described in our submission. The claim of a "fundamental error" is unfounded.
> > >
> > > Our paper includes extensive numerical experiments on nontrivial inverse problems—including motion blur and JPEG compression—all of which produce correct reconstructions. These are not cherry-picked examples; they are standard benchmarks in the generative modeling literature on inverse problems. If our approach contained a fundamental error as claimed, these experiments would fail. They do not.

---

> > > > ### Comment · Reviewer_L5cg · 2025-11-27
> > > > **To clarify**
> > > >
> > > > The embedding is not a nice theoretical issue. It is a must!
> > > > Experiments done only on F:X->X
> > > > Inverse problems are almost always compact which makes the theory and practice different.
> > > >
> > > > Given that the core iterative method relies on an equation (Eq. 8) that is only valid under a restricted assumption, and that the mechanism to overcome that restriction is neither fully defined nor demonstrated, this is a severe point of weakness.
> > > > I retain my score

---

> > > > > ### Author Response · Authors · 2025-12-03
> > > > > **Clarification via a Direct Experimental Demonstration**
> > > > >
> > > > > We respectfully but firmly disagree with the reviewer’s characterization. The assumption underlying Eq. (8) is not a restriction; rather, the critique stems from a misunderstanding of the formulation. As explicitly stated in the first two paragraphs of Section 2.2, our framework addresses inversion at the distribution level, not the recovery of individual latent variables. At this level, it is always possible to introduce an equivalent forward model whose input and output lie within a common ambient space. This construction imposes no conditions on the original inverse problem, and therefore Eq. (8) applies without exception. The embedding step is a standard, well-defined reduction, not an additional theoretical assumption.
> > > > >
> > > > >
> > > > > To remove any remaining ambiguity, we have added a concrete example in Appendix D.4 In this example from the earlier work, the original forward model maps $\mathbb{R}^5$ to $\mathbb{R}^2$, and thus does not satisfy the identity-domain structure. We show two natural embeddings that produce a valid equivalent forward model in a common space and demonstrate empirically that our self-consistent stochastic interpolants operate exactly as intended. Both approaches recover the low-dimensional structure of the underlying distribution, confirming that the method handles settings where $\mathcal{F}$ does not map $\mathcal{X}$ to itself.
> > > > >
> > > > >
> > > > > Taken together with the clarification already present in Section 2.2, this example directly addresses the reviewer’s concern and shows that the claimed limitation does not apply.

---

### Official Review · Reviewer_RjDz · 2025-11-01

**Soundness:** 3
**Presentation:** 3
**Contribution:** 3
**Rating:** 8
**Confidence:** 3

**Summary:**

The authors propose a method to solve the inverse problems, formulated as a generative modeling problem, but in an unusual setup. While usually, when solving the inverse problem, one knows the *clean* data $x$ and corrupt operator $\mathcal{K}$, the problem setup authors solve when one knows the *corrupted* data $y$ and corrupt operator $\mathcal{K}$. Since this setup doesn't have clean data samples, it doesn't allow for the straight application of already developed generative modeling methodologies and requires the development of new ones.

The authors propose to utilize the Stochastic Interpolants (SI) framework and an iterative  (Expectation Maximization-like) scheme for learning it in a no-clean data setup. The key idea of the method is to build such an SI that the corrupted data distribution $Y$ mapped by SI into clean generated data would be mapped back by the corruption operator $\mathcal{K}$ to the initial corrupted data distribution $Y$, i.e., self-consistency property, see lines 177-182. The practical algorithm consists of the iterative repetition of: 1) sampling data from the current SI ($\Theta^{(k)}$), i.e., Expectation step 2) learning the next SI ($\Theta^{(k+1)}$) on the data generated in the previous step, i.e., Minimization step.

In addition, the authors provide a theoretical analysis of their iterative procedure under different conditions and provide convergence guarantees of their procedure.

In practice, the authors evaluate their model on a range of tasks, including toy 2D problems, image inverse problems, and quasar spectrum restoration, and provide comparisons with competing approaches within the image inverse problem setting.

**Strengths:**

- The SI framework allows for a corruption operator $\mathcal{K}$ with black box access, which is less restrictive than previous works  and allows for wider range of corruption operators $\mathcal{K}$.
- The SI is, in general, known to be a good generative modeling framework for the inverse problem solving [1, 2], so the application of SI to such an inverse problem is very well motivated even in $\mathcal{K}$  cases, e.g. linear inverse problems, where one can utilize other diffusion methodologies [3, 4]
- The theoretical analysis is rather comprehensive and convincing. Section 4 results deliver valuable theoretical analysis on the convergence of the method and the possible room for error, which can be helpful in practice and is good as the theoretical result by itself.
- In the image inverse problems experiments method slightly outperforms DPS method that learns *using the clean data* and outperforms other methods that learn only from corrupted data.

**Weaknesses:**

- Some results in the paper are made under the assumption that the corruption operator is *injective*. That seems as not a mild assumption. Can authors comment on the restrictions of such an assumption on practice and show some examples of injective/non-injective mappings?
- The learning procedure is non-simulation free, since one has to sample from learned SI during the "Expectation"-step. That leads to a computationally heavy algorithm. Can authors provide an ablation study on the parameters that control the computational demand of the procedure, i.e., the number of procedure iterations $K$ and the number of gradient updates  $T_{tr}$ inside of one iteration.
- The authors explain the *trade-off* between the condition number $\mathcal{X}$ and restriction of the SI approximation class $\mathcal{S}_\lambda$. This trade-off is shown in Theorem 1, but as far as I understand, authors do not test this trade-off in practice. So, can authors implement the restrictions on SI approximation class in practice and indeed show 1) faster convergence, 2) bigger error. That trade-off and the way to enforce it could be very useful in practice and seems like a useful hyperparameter that hasn't been explored.
- The method's performance gap with EM posterior in Table 2 is marginal and the comparison of the proposed method with other methods that learn only on corrupted data has been carried out on the generation task. Which is strange, as far as I understand, at least some of the reference methods (Ambient Diffusion, EM Posterior or others) can solve inverse problems by starting from corrupted observations. It would be nice to compare the proposed method to competitors in particularly in the image restoration setup and not the generation setup, i.e., add methods that learn only on corrupted data in Table 1.
- The applications of the proposed method are not described properly. Can authors describe the possible real-world scenarios where one has only corrupted observations and a black-box corrupt operator?


[1] Liu, G. H., Vahdat, A., Huang, D. A., Theodorou, E., Nie, W., & Anandkumar, A. (2023, July). I $^ 2$ SB: Image-to-Image Schrödinger Bridge. In _International Conference on Machine Learning_ (pp. 22042-22062). PMLR.

[2] Albergo, M. S., Goldstein, M., Boffi, N. M., Ranganath, R., & Vanden-Eijnden, E. (2024, July). Stochastic Interpolants with Data-Dependent Couplings. In _International Conference on Machine Learning_ (pp. 921-937). PMLR

[3] Francois Rozet, Gerome Andry, Franc¸ois Lanusse, and Gilles Louppe. Learning diffusion priors from
observations by expectation maximization. Advances in Neural Information Processing Systems,
37:87647–87682, 2024.

[4] Giannis Daras, Kulin Shah, Yuval Dagan, Aravind Gollakota, Alex Dimakis, and Adam Klivans. Ambient diffusion: Learning clean distributions from corrupted data. Advances in Neural Information
Processing Systems, 36:288–313, 2023.

**Questions:**

See Weaknesses section

---

> ### Author Response · Authors · 2025-11-22
> **Response 1/2**
>
> We thank you for your careful and thorough review, for recognizing the strengths of our method, and for providing many valuable suggestions. Below, we provide our responses to your questions.
>
> **Q1.** Assumption on the injectivity of the forward operator at the distribution level.
>
> **A.** Thanks for this important comment, it touches on a key structural property of the problem. Crucially, injectivity here is defined at the level of the induced operator on *probability measures*, not at the level of the sample-wise corruption map. Even when the map $x \mapsto \mathcal{F}(x) $ is non-injective (e.g., a projection losing dimensions), the operator $\mathcal{K}$ acting on distributions can still be injective, meaning that $\mathcal{K}\pi_1 = \mathcal{K}\pi_2 $​ implies $\pi_1 = \pi_2 $. In essence, an injective corruption operator ensures that the input data distribution can be uniquely recovered from the output data distribution.
>
> Some examples of injective corruption operators include:
> - The AWGN channel, where $\mathcal{F}(x) = x + \sigma w$, and $w \sim N(0, I)$, for any $\sigma < \infty$. This is because at the level of distributions, the channel is a Gaussian convolution: $\mu = \pi * \gamma_{\sigma}$, which is injective for any finite variance. This extends to any additive non-Gaussian channel, provided the noise distribution has finite moments of any order.
> - The ‘random projection’ channel, where $\mathcal{F}(x) = (A^T x, A)$, and $A$ is drawn independently of $x$ from a distribution $\rho$ over unitary matrices $A$ in $\mathbb{R}^{d \times r}$ , $r < d$ (the so-called Stiefel manifold) such that $\text{span} (A; A \text{ in the support of }\rho) = \mathbb{R}^d$. In words, we randomly project the input $x$ to a low-dimensional feature $Ax$, in such a way that any input direction has a positive probability of being observed eventually. Examples include the inpainting channel with random masks and tomography.
> - The ‘erasure’ channel, where $\mathcal{F}(x) = x$ with probability $\epsilon>0$, and $\mathcal{F}(x) = 0$ otherwise. Even though most samples will be thrown, at the level of distributions, the operator is $\mathcal{K}\pi = \epsilon \pi + (1-\epsilon) \delta_0$, which is invertible.
>
> Some examples of non-injective operators include:
> - The ‘blind’ projection channel $\mathcal{F}(x) = A^T x$, where we do not observe the range $A \in \mathbb{R}^{d \times r}$ associated with each measurement. Indeed, here even if $r=d$ the channel can fail to be injective. For example, if $A$ is drawn from the uniform measure over the unitary group (Stiefel when $r=d$), the channel is rotation invariant, so if the original $\pi$ is not rotation invariant, i.e., $\pi \neq R_{\\#} \pi$ for some rotation $R$, we have that $\mathcal{K} R_{\\#} \pi = \mathcal{K}\pi$, so the channel cannot distinguish $\pi$ from $R_{\\#} \pi$.
>
> - The JPEG channel using only a fixed quality level $q < 1$ is likely not injective, since high-frequency information will be systematically lost by the quantization.
>
> **Q2.** An ablation study on the parameters that control the computational demand of the procedure, i.e., the number of procedure iterations  and the number of gradient updates  inside of one iteration.
>
> **A.** Thanks for the comment. We do agree that the hyperprameter $T_{\text{tr}}$ influences the computational cost of the procedure. In Appendix C.1 (the bottom of page 17 of the original PDF), we explained the practical choice of that as follows:
> > We experimented with different choices of $T_{\text{tr}}$, the number of backward transport steps in Alg. 1, and observed only minor differences across values, with $T_{\text{tr}} = 1$ already sufficient for all current experiments. For simplicity, we therefore set $T_{\text{tr}} = 1$ throughout this work.
>
> To illustrate this more concretely, we revisited the experiment in Section 5.1 (Fig. 2) and varied $T_{\text{tr}}$ while keeping the product $T_{\text{tr}} \cdot K$ fixed, so the total computational budget remained the same. The Wasserstein distance between the final solved distribution and the true distribution is reported in the table below (with three independent runs). The results indicate that the performance is quite insensitive to $T_{\text{tr}}$ as long as it is not chosen so large that it forces too few outer iterations. In addition to the final performance, we also do not find any significant differences in the convergence speeds. Based on this robustness and to avoid unnecessary hyperparameter tuning, we use $T_{\text{tr}} = 1$ in all our experiments.
>
> | $T_{\text{tr}}$ | Wasserstein Distance (Mean ± Std) |
> |:---------------:|:---------------------------------:|
> |         1       | 0.0491 ± 0.0038                   |
> |        10       | 0.0460 ± 0.0038                   |
> |       100       | 0.0476 ± 0.0047                   |
> |      1000       | 0.0593 ± 0.0021                   |
>
> We will add the ablation results to the appendix in the updated version for completeness.

---

> > ### Author Response · Authors · 2025-11-22
> > **Response 2/2**
> >
> > **Q3.** Theorem 1: Trade-off between the SI approximation class and convergence rate w.r.t. $k$.
> >
> > **A.**  Thanks for this question. It raises an interesting point, although the situation is more nuanced in practice. The error term $\delta^{(k)} := \max( \| b^{(k)} - \hat{b}^{(k)} \|, \| s^{(k)} - \hat{s}^{(k)} \|)$ at each outer iteration $k$ depends on both the SI approximation class *and* the optimization quality. Our discussion of the trade-off in the manuscript focuses on the approximation aspect and implicitly assumes that the optimization error is reasonably controlled. In practical training, however, we choose the number of optimization steps to be small in each outer iteration $k$ (as explained in the previous point) to maintain overall computational efficiency. Under this regime, the optimization accuracy becomes a significant factor, and the theory does not fully capture this computation–accuracy trade-off. Consequently, varying the approximation class in isolation would not yield a clean or meaningful illustration of the theoretical trade-off, and the proposed experiment would not reflect the behavior of the method in practice. We will revise the manuscript to clarify this point.
> >
> > **Q4.** Comparison in the individual image restoration setup.
> >
> > **A.** The reviewer is correct that methods trained only on corrupted data, such as EM Posterior, can in principle be used for inverse problems (i.e., restoring samples). We did not include them in the image restoration setup because we did not reimplement these methods and instead reported their results in the generative setting as presented in their original papers. Reproducing EM Posterior in particular is computationally demanding (reported to require around 512 GPU hours, compared to our 86). In addition, both Ambient Diffusion and EM Posterior were evaluated only on random masking in their original work and are not applicable to the nonlinear motion-blur corruption used in Table 1. For these reasons, we chose to include only DPS as a reference method, noting that it is solved in a different and easier setting.
> >
> > **Q5.** Further explanation of the real-world applications.
> >
> > **A.** We will make this more explicit in the revised version, but the last two experiments (JPEG restoration and quasar spectra) were meant to exemplify the real-world scenarios. In experimental sciences, it is often the case that we only have access to observations while we are interested in the underlying clean data - e.g., we only observe astronomical objects like quasars, pulsars etc. through a telescope; background radiation through intervening dust in the Universe. In medical sciences, MRI often does k-space subsampling to reduce scan times, and similarly for Cryo-EM.

---

> > > ### Comment · Reviewer_RjDz · 2025-11-25
> > > **Thank you for answering my questions**
> > >
> > > I thank the authors for the detailed answer to my weaknesses/questions. Please incorporate the corresponding changes to the paper revision. I will keep my original score.

---

### Author Response · Authors · 2025-11-26
**Summary of revisions and additional results**

We sincerely thank all four reviewers for their time and valuable comments. We appreciate reviewers RjDz, vQVs, and hPBy for recognizing the strengths of our work. We must also respectfully reiterate that the central criticism raised by reviewer L5cg—namely, that our path definition is fundamentally flawed because $\mathcal{F}(x)$ and $x$ live in different spaces—reflects a misreading of our paper. The score of 0 (strong reject) with confidence 5 (absolutely certain) is therefore not only disproportionate to the actual content of our work, but also at odds with the assessments of the three other reviewers who assigned scores of 6, 8, and 8. We hope that our response and the corresponding revisions help resolve this misunderstanding.

Our point-by-point responses address all concerns and questions in detail, with additional clarification and expanded experimental evidence. We have also uploaded a revised manuscript incorporating all changes discussed in the rebuttal. Modifications and additions are highlighted in blue. The main updates are:

- At the beginning of Section 2, we further clarify why one can always work with a common space for both the input and output of the simulator $\mathcal{F}$, either formally through a product space or, in many practical settings, via an embedding.

- In the middle of section 2.2, we clarify that our injectivity assumption is imposed at the distribution level as a basic assumption of the problem, and we provide concrete examples where this assumption holds.

- In the introduction, we add further motivation for our central task—an inverse problem at the distribution level, or inverse generative modeling. At the end of Section 2.2, we clarify the distinctions between restoration, generation, and inference, so as to better position our focus on generation rather than sample-level restoration.

- In Appendix D.3, we include additional metrics for the individual restoration task, showing that our method compares favorably to DPS, even though DPS is trained on clean data while our method is not. More importantly, we report results from an oracle model trained on paired clean–corrupted data, which serves as an upper bound. Our method remains reasonably close to this oracle across most tasks.

- In Appendix D.2, we provide additional SDE-based results for both imaging and quasar experiments. We observe that in high-dimensional settings, performance of SDE is more sensitive to hyperparameters such as the noise schedule and the number of transport steps. By contrast, the ODE approach performs well under moderate corruptions, is robust to hyperparameter choices, and is less computationally expensive. Due to limited computational budget, we have not extensively tuned SDE parameters, and the current results are therefore comparable to or slightly weaker than those in the ODE setting. This warrants further investigation.

- In Appendix C.2, we present additional ablations supporting the choice $T_{\text{tr}} = 1$ across all experiments. Appendix D.1 includes further ablations on the effect of network size on performance.

- For the quasar example, we introduce a Wiener-filter baseline and show that our method substantially outperforms it in both considered settings.

Separately, we would also like to report one additional experiment that completed only after we prepared our point-by-point responses. Because this setting is expensive to run and represents only a single configuration, we do not plan to add it to the revised manuscript, but include it here for completeness. This concerns the individual restoration baseline raised by reviewers RjDz and hPBy. We implemented the EM Posterior method ourselves (reported to require around 512 GPU hours, compared to our 86) for the random masking task with 50% corruption and low noise (the first row of Table 1). Using the trained model to perform restoration as in Table 1 and Table 6, the EM Posterior results for LPIPS $\downarrow$ / PSNR $\uparrow$ / SSIM $\uparrow$ are 0.00613 / 28.60 / 0.936, whereas our method achieves 0.00503 / 28.97 / 0.931. Although restoration is not the primary focus of our work, as clarified in the revision, this comparison indicates that our method remains slightly stronger than this existing inverse generative modeling approach while requiring far less computation.

We thank the reviewers once again for their thorough review of our work and look forward to any further feedback.

---

### Author Response · Authors · 2025-12-03
**Summary for the new AC**

Dear Area Chair,

We thank the new AC for graciously looking at our submission in light of the recent situation. We would like to summarize our discussion with the reviewers in this message. Our changes to the manuscript in response to these discussions are highlighted in ‘blue’ color for convenience.

With reviewers RjDz and vQVs, we had a productive discussion and were able to satisfactorily answer their questions as well as provide clarifications. Both reviewers reconfirmed their high score of 8.

Regarding reviewer hPBy: the reviewer had asked use for several new experiments- i) SDE runs for the high dimensional  examples, ii) new metrics to compare with DPS method on restored samples, iii) additional baselines for restoration task and iv) quantitative metrics as well as a  baseline for quasar example. We have provided these in the revised manuscript and the reviewer seemed satisfied with these.
They had one follow-up question which requested for quantitative evaluation of SDE results in addition to the qualitative results provided in the first response. We have not included these in the revised manuscript yet due to computational constraints of running these large experiments in the rebuttal period. However we have these results for ablation exercises on small networks to show that SDE can perform comparably to ODE for some hyperparameter settings. These numbers are now provided in our latest response to the reviewer.
In addition, the reviewer’s original message referred to the inconsistency that we only had theory for SDE while most experiments were done for ODE setting. To address this, we have also added new theoretical results discussing contraction for both ODE and SDE approaches, bringing our theory and experiments into closer alignment.

Regarding Reviewer L5cg who gave us a score of 0, we remain in disagreement with their assessment. Their objection is that we cannot define an interpolant when the data and observations do not lie in the same space. This is something which we had already discussed in the previous manuscript by appealing to padding and product embeddings, and have elaborated on this in our responses to the reviewer. In the most recent version of the manuscript, we have also added a new example in the appendix that explicitly considers such an example and demonstrates the applicability of our method.

---

### Meta-Review · Area_Chair_AXDk · 2026-01-14

**Summary:**

This paper makes interesting progress on solving inverse problems when given only access to noisy data (the ambient diffusion problem).
The authors propose to utilize the Stochastic Interpolants (SI) framework and an iterative EM algorithm. The paper is interesting, novel and well-executed.

**Reviewer Concerns:**

Reviewer L5cg assigned zero in a very short review that has a valid concern but then that was addressed (I think) by the reviewers. After reading the discussion I think the authors addressed the comment and the zero score was not warranted.

**Reviewer Scores:**

I think one review was unfair and I discount it.

---

### Decision · Program_Chairs · 2026-01-26

Accept (Poster)